# T-GINEE: A Tensor-Based Multilayer Graph Representation Learning

**Maolin Wang** [1]  **Ziting Mai** [1]  **Xuhui Chen** [1]  **Zhiqi Li** [1 2]  **Tianshuo Wei** [1]  **Yutian Xiao** [3]  **Wenlin Zhang** [1]
**Wanyu Wang** [1]  **Ruocheng Guo** [4]  **Haoxuan Li** [5]  **Zenglin Xu** [6 2]  **Xiangyu Zhao** [1]

## Abstract

While traditional network analysis focuses on single-layer networks, real-world systems often form multilayer networks with multiple relationship types. However, existing methods typically fail to capture complex inter-layer dependencies by treating layers independently or aggregating them. To address this, we propose T-GINEE (Tensor-Based Generalized Multilayer-graph Estimating Equation), a statistical regularization framework combining tensor-based generalized estimating equations with task-specific loss to model cross-network correlations explicitly. Key innovations include: (1) CP tensor decomposition capturing structural dependencies via shared latent factors; (2) a generalized estimating equation framework modeling inter-layer correlations through working covariance matrices; and (3) a flexible link function accommodating characteristics like sparsity. Our theoretical analysis establishes consistency and asymptotic normality under mild conditions. Extensive experiments on synthetic and real-world datasets validate T-GINEE's effectiveness for multilayer network analysis. Our code is available at https://github.com/Applied-Machine-Learning-Lab/ICML2026_T-GINEE.

## 1. Introduction

In the real world, interactions between entities are often multifaceted, with these multi-relational characteristics engaging one another under varied circumstances or through distinct modalities. For instance, in social networks (Van Den Oord & Van Rossem, 2002), individuals may be connected through multiple relationship types such as friends, colleagues, and family. In biology (Zheng et al., 2019), genes or proteins exhibit various collaboration schemes like co-expression and physical interactions. In global trade, countries exchange a wide range of different commodities.

For such intricate relational landscapes, a multi-layer graph offers a faithful and structured representation. This architecture is defined by a common set of vertices, where each layer is endowed with a unique edge set to delineate a specific type of relation. Such graphs are prevalent across numerous disciplines, including social graphs that capture multiple interaction channels between individuals (Greene & Cunningham, 2013), biological graphs detailing different collaboration schemes among genes or proteins (Li et al., 2020; Liu et al., 2020), and global trade graphs mapping the exchange of various commodities (Alves et al., 2019; Ren et al., 2020). To effectively analyze these intricate structures, a fundamental step is to learn low-dimensional vector representations (i.e., embeddings) for the entities that capture the complex relational information encoded across layers.

Numerous approaches have been developed for graph embedding, employing various techniques such as similarity indices (Boden et al., 2017), maximum likelihood models (Yuan & Qu, 2021), matrix factorization (Tang et al., 2009; Dong et al., 2012; Gligorijević et al., 2016), and graph neural networks (Kipf & Welling, 2016; Hamilton et al., 2017; Xu et al., 2018). For multilayer graph embedding, which provides a richer representation of complex systems (Kivelä et al., 2014), analysis often involves extending these single-layer techniques. Prominent approaches include tensor-based methods that leverage the natural tensor structure of multilayer graphs (Kolda & Bader, 2009; Aguiar et al., 2024), as well as adaptations of deep learning models like GCNs and random-walk embeddings (Ghorbani et al., 2019; Song & Thiagarajan, 2018).

However, a critical challenge underlying many of these methods is the lack of a rigorous theoretical foundation for the multi-layer context. While embedding learning has proven effective for single-layer graphs (Cai et al., 2018; Wang et al., 2023b), we lack robust theoretical frameworks systematically characterizing the embedding process across multiple layers (Interdonato et al., 2020). This absence of formal tools describing how embeddings capture and

---
[1]City University of Hong Kong, Hong Kong SAR, China [2]Shanghai Academy of AI for Science, Shanghai, China [3]Beihang University, Beijing, China [4]Independent Researcher [5]Peking University, Beijing, China [6]Fudan University, Shanghai, China. Correspondence to: Xiangyu Zhao <xianzhao@cityu.edu.hk>.

*Proceedings of the $43^{rd}$ International Conference on Machine Learning*, Seoul, South Korea. PMLR 306, 2026. Copyright 2026 by the author(s).

preserve cross-layer dynamics significantly impedes developing principled approaches, representing a fundamental limitation in the field (Shanthamallu et al., 2019; Jiao et al., 2021; Lyu et al., 2023).

Without this theoretical guidance, existing approaches often resort to simplistic solutions, such as learning representations for layers independently (Tang et al., 2009; Dong et al., 2012) or using basic aggregation techniques (Paul & Chen, 2020; Lei et al., 2020). These methods lack the grounding to explain how embeddings should optimally encode the nuanced ways in which relationships in one layer might influence or contradict another (Liu et al., 2017; Zhang et al., 2018). This deficit is especially problematic for real-world systems where entities engage through multiple relation types simultaneously (Xu et al., 2020; Yang et al., 2020), highlighting the urgent need for new frameworks that can faithfully represent this complex interplay (Wang et al., 2024; Huang et al., 2020; Shanthamallu et al., 2019).

To address these challenges, we propose T-GINEE (Tensor-based Generalized Multilayer-graph Estimating Equation), a statistical regularization framework that combines tensor-based generalized estimating equations with task-specific loss to explicitly model cross-network correlations. The key technical innovations of T-GINEE include: (1) A CP tensor decomposition approach that effectively captures structural dependencies through shared latent factors while maintaining computational efficiency; (2) A generalized estimating equation framework that explicitly models the correlations between different network layers through working covariance matrices; and (3) A flexible link function design that accommodates various network characteristics, including sparsity. Unlike previous approaches that rely on simple aggregation or separate modeling (Paul & Chen, 2020; Tang et al., 2009; Lei et al., 2020), T-GINEE provides a principled statistical framework to jointly model multiple networks.

Overall, T-GINEE integrates a symmetric CP tensor decomposition with a generalized estimating equation (GEE) formulation, provides asymptotic statistical guarantees, and validates the framework on both synthetic and real-world multilayer networks. The main contributions are summarized as follows:

- **Tensor-based Statistical Framework**: We propose a regularization framework combining tensor CP decomposition with generalized estimating equations. This approach explicitly models cross-network dependencies via a principled formulation while ensuring tractability.

- **Theoretical Guarantees**: We establish T-GINEE's consistency and asymptotic normality for the full-batch estimating-equation formulation under explicit regularity and dimensional-growth assumptions. These results clarify when the tensor-based score equations provide

statistically reliable estimation.

- **Empirical Validation**: Comprehensive experiments on synthetic and real-world networks demonstrate T-GINEE's effectiveness, including link prediction, community detection, GNN comparisons, covariance ablations, robustness checks, and large-scale profiling.

## 2. Methodology

In this section, we present Tensor-based Generalized Estimating Equations (T-GINEE), a framework for learning embeddings from multilayer graphs. Our method combines a low-rank CP parameterization of a multilayer graph with a generalized estimating equations (GEE) estimator, equipped with a structured working covariance, to jointly model within-layer and cross-layer dependencies.

### 2.1. Overview

Real-world networks often exhibit complex interdependencies, where multiple network structures coexist and influence each other. For instance, an individual's friendship networks on multiple social media platforms (such as Facebook, LinkedIn, and TikTok) form correlated multilayer graphs over the same set of users. We propose a statistical regularization framework that leverages tensor-based generalized estimating equations to explicitly model such cross-network correlations. Our framework, referred to as T-GINEE, consists of several core components illustrated in Figure 1: (i) a symmetric CP decomposition of the parameter tensor $\Theta$ of the multilayer graph into node embeddings $\alpha$ and layer-specific embeddings $\beta$; (ii) the construction of a parameter vector $\gamma$ from these embeddings; and (iii) a tensor-based GEE formulation that estimates $\gamma$ under a working covariance model and thereby captures complex dependencies across layers.

### 2.2. Preliminaries and notation

We briefly introduce the tensor and estimating-equation notation used below. For vectors $a \in \mathbb{R}^n$ and $b \in \mathbb{R}^m$, $a \otimes b$ denotes the Kronecker product. For matrices $A = [a^{(1)}, \ldots, a^{(R)}]$ and $B = [b^{(1)}, \ldots, b^{(R)}]$ with the same number of columns, $A \odot_{\mathrm{KR}} B = [a^{(1)} \otimes b^{(1)}, \ldots, a^{(R)} \otimes b^{(R)}]$ denotes the Khatri–Rao product. A rank-$R$ CP decomposition writes a third-order tensor as a sum of rank-one outer products. In our symmetric multilayer setting, this takes the form $\Theta = \sum_{r=1}^{R} \alpha^{(r)} \circ \alpha^{(r)} \circ \beta^{(r)}$, where $\alpha$ contains node factors and $\beta$ contains layer factors. The GEE component treats $\mathcal{A}_{i,j,\cdot}$ as an $M$-dimensional response for node pair $(i, j)$ and weights its residual by a working covariance matrix. This lets T-GINEE model cross-layer residual dependence without specifying a full joint distribution over the entire adjacency tensor.

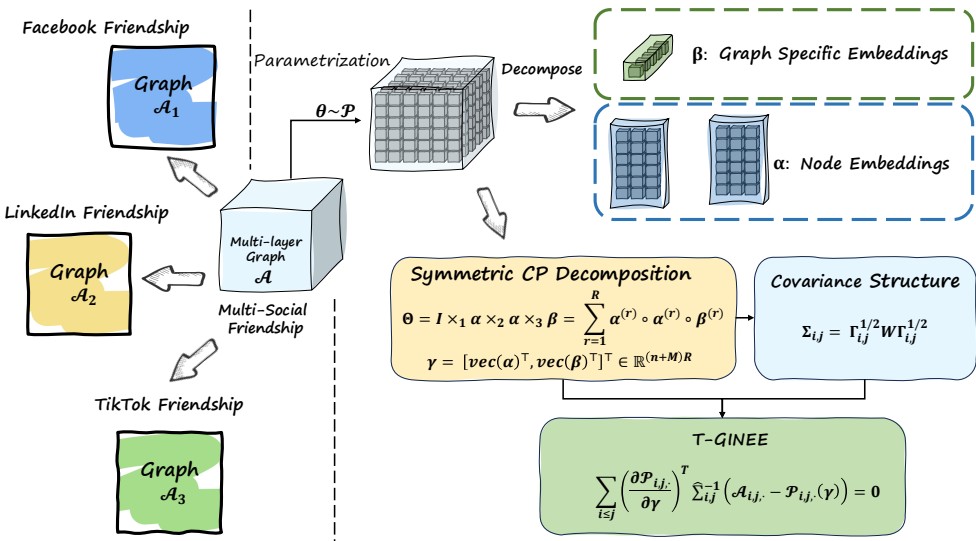

*Figure 1.* Schematic overview of the T-GINEE framework. Given a multilayer adjacency tensor $\mathcal{A}$, we introduce a parameter tensor $\Theta$ and perform a symmetric CP decomposition to obtain node embeddings $\alpha$ and layer-specific embeddings $\beta$. These embeddings are concatenated into a parameter vector $\gamma$. By solving tensor-based generalized estimating equations (T-GINEE) under a working covariance structure, the model learns $\gamma$ and captures cross-layer dependencies in multilayer social networks (e.g., Facebook, LinkedIn, TikTok).

## 2.3. Problem formulation

Consider a multilayer network/graph $\mathcal{G} = (\mathcal{V}, \{\mathcal{G}^{(m)}\}_{m=1}^{M})$, where $\mathcal{V} = \{v_1, \ldots, v_n\}$ is a common set of vertices that interact across $M$ different but potentially correlated network layers. Each layer $\mathcal{G}^{(m)} = (\mathcal{V}, \mathcal{E}^{(m)})$ captures a distinct type of relationship. We focus on undirected networks and index undirected edges by unordered pairs $\{i, j\}$, using the convention that $i \leq j$. In our notation we include pairs with $i = j$; in the empirical datasets we consider, the diagonal entries are identically zero (i.e., $\mathcal{A}_{i,i,m} \equiv 0$), so allowing $i = j$ does not affect estimation but simplifies asymptotic analysis.

Let $\mathcal{A} \in \{0,1\}^{n \times n \times M}$ be the adjacency tensor, where $\mathcal{A}_{i,j,m} = 1$ indicates an edge of type $m$ between nodes $i$ and $j$, and $\mathcal{A}_{i,j,m} = 0$ otherwise. For each pair $(i,j)$ with $i \leq j$, the vector $\mathcal{A}_{i,j,\cdot} \in \mathbb{R}^M$ is treated as an $M$-dimensional binary response with mean $\mathcal{P}_{i,j,\cdot}(\Theta)$ and covariance matrix $\Sigma_{i,j}$. In line with the GEE paradigm, we specify only its first two moments, that is $\mathbb{E}[\mathcal{A}_{i,j,\cdot}] = \mathcal{P}_{i,j,\cdot}(\Theta)$ and $\mathrm{Cov}(\mathcal{A}_{i,j,\cdot}) = \Sigma_{i,j}$, without imposing a full joint distribution. Conditional on the parameters, we assume that the edge vectors $\{\mathcal{A}_{i,j,\cdot} : i \leq j\}$ are independent across different node pairs $(i,j)$. Within each pair $(i,j)$, the components $\mathcal{A}_{i,j,1}, \ldots, \mathcal{A}_{i,j,M}$ may be correlated and this dependence is captured by $\Sigma_{i,j}$.

The parameter tensor $\Theta \in \mathbb{R}^{n \times n \times M}$ is linked to the mean tensor $\mathcal{P}$ through a known three-times continuously differentiable link function $g$, applied elementwise, such that

$$\mathcal{P}_{i,j,\cdot} = g^{-1}(\Theta_{i,j,\cdot}).$$

Typical choices of the link function include:

- the identity link $g(x) = x$, primarily used for weighted networks where $\mathcal{A}_{i,j,m}$ need not be binary; for Bernoulli observations, the identity link can be applied by implicitly constraining $\Theta_{i,j,m} \in [0,1]$;

- the probit link $g(x) = \Phi^{-1}(x)$ with inverse $g^{-1}(x) = \Phi(x) = \int_{-\infty}^{x} (2\pi)^{-1/2} \exp(-t^2/2) \, \mathrm{d}t$;

- the logit link $g(x) = \log\{x/(1-x)\}$ with inverse $g^{-1}(x) = 1/(1 + e^{-x})$;

- a sparsity-aware logit $g(x) = \log\{x/(s-x)\}$ with inverse $g^{-1}(x) = \dfrac{s}{1 + e^{-x}}$, where $0 < s < 1$ is a sparsity coefficient that can decrease with $n$ and $M$ to accommodate increasingly sparse networks. When $s$ is allowed to depend on $(n, M)$, we assume there exists $\varepsilon > 0$ such that, for all $(i,j,m)$ and all $\gamma$ in a neighborhood of $\gamma_0$,

$$\varepsilon \leq \mathcal{P}_{i,j,m}(\gamma) \leq s - \varepsilon,$$

which in particular guarantees that $g'$ and the required higher-order derivatives remain uniformly bounded, in line with Assumption 3.1.

## 2.4. Low-rank tensor decomposition

Before introducing the formal CP decomposition (Wang et al., 2023a; Kolda & Bader, 2009), we briefly motivate this modeling choice. In many multilayer systems, the same set of nodes participates in different relation types (layers)

through a shared, low-dimensional set of latent traits (e.g., social roles, functional modules, or economic profiles). A symmetric CP factorization of the parameter tensor $\Theta$ reflects this idea: the node embeddings $\alpha$ define a common latent space across all layers, while the layer embeddings $\beta$ determine how each relation type weights these latent factors. This yields a compact representation that captures higher-order interactions across nodes and layers, reduces the number of free parameters, and aligns with empirical findings that a relatively small number of latent dimensions often suffices to explain multilayer network structure.

The formal asymptotic analysis in Section 3 is stated for the full-batch estimating-equation regime under explicit dimensional-growth assumptions. The scalable mini-batch implementation used for massive graphs is described separately in Appendix C.

To effectively model structural dependencies while maintaining computational efficiency, we employ a symmetric CP decomposition for the parameter tensor $\Theta$:

$$\Theta = \mathcal{I} \times_1 \alpha \times_2 \alpha \times_3 \beta = \sum_{r=1}^{R} \alpha^{(r)} \circ \alpha^{(r)} \circ \beta^{(r)}, \quad (1)$$

where $\mathcal{I} \in \mathbb{R}^{R \times R \times R}$ denotes the order-3 identity tensor used in CP decomposition, $\alpha \in \mathbb{R}^{n \times R}$ contains node embeddings and $\beta \in \mathbb{R}^{M \times R}$ contains layer-specific embeddings. Here $\alpha^{(r)}$ and $\beta^{(r)}$ denote the $r$-th columns of $\alpha$ and $\beta$, respectively, and $\circ$ denotes the outer product. This parameterization enforces $\Theta_{i,j,m} = \Theta_{j,i,m}$ for all $i, j, m$, consistent with undirected layers, and inherently imposes structural constraints through shared latent factors.

For optimization purposes, we vectorize these factor matrices into a compact representation:

$$\gamma = \left[ \mathrm{vec}(\alpha)^{\top}, \ \mathrm{vec}(\beta)^{\top} \right]^{\top} \in \mathbb{R}^{(n+M)R}. \quad (2)$$

This parameter vector $\gamma$ encapsulates the essential cross-layer dependencies and serves as the decision variable in our estimating equations. The low-rank tensor decomposition offers several advantages: it significantly reduces the number of free parameters, improving computational efficiency and mitigating overfitting; by sharing latent factors across dimensions, it naturally captures the inherent relationships between nodes and layers; and it yields interpretable components, where $\alpha$ represents node-level patterns and $\beta$ captures layer-level characteristics.

### 2.5. Tensor-based statistical regularization

We now introduce the tensor-based generalized estimating equations that define T-GINEE. A detailed derivation is provided in **Appendix** A. The tensor-based estimating equa-

tions for multilayer graphs are

$$\sum_{i \leq j} \left( \frac{\partial \mathcal{P}_{i,j,\cdot}}{\partial \gamma} \right)^{\top} \widehat{\Sigma}_{i,j}^{-1} \left( \mathcal{A}_{i,j,\cdot} - \mathcal{P}_{i,j,\cdot}(\gamma) \right) = \mathbf{0}, \quad (3)$$

where $\widehat{\Sigma}_{i,j}$ is a working covariance matrix. We denote the left-hand side of (3) by $s(\gamma)$; thus T-GINEE seeks a root $\hat{\gamma}$ of the estimating equation $s(\gamma) = \mathbf{0}$.

To compute the score contributions, we first derive $\partial \mathcal{P}_{i,j,\cdot}/\partial \gamma$ via the chain rule: starting from the Jacobian $\partial \mathrm{vec}(\Theta)/\partial \gamma$ implied by the CP decomposition, then applying the derivative of the link function, and finally projecting onto the $(i,j)$-th fibers using basis tensors $\mathcal{E}^{(i,j,m)}$.

Specifically, since $\mathcal{P}_{i,j,m} = g^{-1}(\Theta_{i,j,m})$ and $g$ is differentiable, we have

$$\frac{\partial \mathcal{P}_{i,j,\cdot}}{\partial \gamma} = \left[ \mathrm{diag}\left( g'(\mathcal{P}_{i,j,1}), \ldots, g'(\mathcal{P}_{i,j,M}) \right) \right]^{-1} \frac{\partial \Theta_{i,j,\cdot}}{\partial \gamma} \quad (4)$$

where $g'$ denotes the derivative of $g$, applied elementwise, and the diagonal matrix has $(m,m)$-th entry $g'(\mathcal{P}_{i,j,m})$ for $m \in [M]$. Let $\mathcal{E}^{(i,j,m)} \in \mathbb{R}^{n \times n \times M}$ be the tensor unit whose $(i', j', m')$-th entry is $\mathbf{1}\{(i,j,m) = (i',j',m')\}$. Then

$$\frac{\partial \Theta_{i,j,\cdot}}{\partial \gamma} = \left[ \mathrm{vec}(\mathcal{E}^{(i,j,1)}), \ldots, \mathrm{vec}(\mathcal{E}^{(i,j,M)}) \right]^{\top} \frac{\partial \mathrm{vec}(\Theta)}{\partial \gamma} \quad (5)$$

Under the CP decomposition of $\Theta$, the Jacobian matrix with respect to the parameter vector $\gamma$ takes the block form

$$\frac{\partial \mathrm{vec}(\Theta)}{\partial \gamma} = \begin{bmatrix} (\beta^{(1)})^{\top} \otimes \left( I_n \otimes \alpha^{(1)} + \alpha^{(1)} \otimes I_n \right) \\ \vdots \\ (\beta^{(R)})^{\top} \otimes \left( I_n \otimes \alpha^{(R)} + \alpha^{(R)} \otimes I_n \right) \\ I_M \otimes \left( \alpha \odot_{\mathrm{KR}} \alpha \right) \end{bmatrix}, \quad (6)$$

where $\otimes$ denotes the Kronecker product, and $\odot_{\mathrm{KR}}$ denotes the Khatri–Rao (column-wise Kronecker) product:

$$\alpha \odot_{\mathrm{KR}} \alpha = \left[ \alpha^{(1)} \otimes \alpha^{(1)}, \ldots, \alpha^{(R)} \otimes \alpha^{(R)} \right] \in \mathbb{R}^{n^2 \times R}.$$

Each of the first $R$ block rows in (6) has dimension $(n^2 M) \times n$, and the last block row $I_M \otimes \left( \alpha \odot_{\mathrm{KR}} \alpha \right)$ has dimension $(n^2 M) \times (MR)$, so that $\partial \mathrm{vec}(\Theta)/\partial \gamma \in \mathbb{R}^{(n^2 M) \times (n+M)R}$. See **Appendix** A for a detailed derivation of $\partial \mathrm{vec}(\Theta)/\partial \gamma$.

Combining (4)–(6) yields the full expression for $\partial \mathcal{P}_{i,j,\cdot}/\partial \gamma$, and the complete formulation is presented in **Appendix** B.

### 2.6. Covariance structure and estimation

The cross-layer dependencies within each node pair $(i,j)$ are summarized by the covariance matrices $\Sigma_{i,j} = \mathrm{Cov}(\mathcal{A}_{i,j,\cdot})$. In line with the GEE framework, we approximate these true covariances by a parsimonious working

covariance family that assumes a common correlation structure across node pairs:

$$\Sigma_{i,j}^{\mathrm{w}}(\gamma) = \Gamma_{i,j}^{1/2} W \Gamma_{i,j}^{1/2}, \tag{7}$$

where $\Gamma_{i,j} \in \mathbb{R}^{M \times M}$ is a diagonal matrix whose $(m,m)$-th entry is $\mathcal{P}_{i,j,m}(\gamma)\big(1 - \mathcal{P}_{i,j,m}(\gamma)\big)$, and $W \in \mathbb{R}^{M \times M}$ is a positive-definite correlation matrix shared across node pairs. The working covariance matrices appearing in (3) are then

$$\widehat{\Sigma}_{i,j} = \Gamma_{i,j}^{1/2} \widehat{W} \Gamma_{i,j}^{1/2},$$

where $\widehat{W}$ is an empirical estimate of $W$.

We estimate $W$ by pooling residuals across all node pairs:

$$\widehat{W} = \tfrac{1}{N} \sum_{i \le j} \Gamma_{i,j}^{-1/2} \big(\mathcal{A}_{i,j,\cdot} - \mathcal{P}_{i,j,\cdot}(\hat\gamma)\big)\big(\mathcal{A}_{i,j,\cdot} - \mathcal{P}_{i,j,\cdot}(\hat\gamma)\big)^{\top} \Gamma_{i,j}^{-1/2} \tag{8}$$

where $\hat\gamma$ is the current estimate of $\gamma$, $N = n(n+1)/2$ is the number of node pairs with $i \le j$, and the sum runs over all such pairs, consistent with our convention in Section 2.4. Under the boundedness and working-covariance assumptions, the eigenvalues of $\Sigma_{i,j}^{\mathrm{w}}(\gamma)$ are bounded away from zero and infinity, which ensures numerical stability and underpins our asymptotic analysis in Section 3. In practice, to avoid numerical instability when inverting $\widehat{\Sigma}_{i,j}$, we may add a small ridge term $\epsilon I_M$ to $\widehat{W}$.

### 2.7. Optimization and computational complexity

In practice, we do not solve the nonlinear estimating equations (3) in closed form. Instead, we optimize $\gamma$ using iterative gradient-based updates, alternating with periodic updates of the working correlation matrix $W$.

**Optimization procedure.** At a high level, each training epoch consists of: (i) computing the CP-based parameter tensor $\Theta$ and the corresponding edge probabilities $\mathcal{P}_{i,j,m}(\gamma)$ for sampled node pairs $(i,j)$ and layers $m$; (ii) evaluating the score contributions in (3) via the Jacobian $\partial \mathcal{P}_{i,j,\cdot}/\partial\gamma$; and (iii) updating $\gamma$ by a gradient step (with optional regularization), followed by an update of $W$ using (8). In our implementation we initialize $W$ as the identity $I_M$ (an independence working structure) and start updating it after a few warm-up epochs.

**Per-iteration complexity and scalability.** In the dense case, computing the CP-based parameter tensor $\Theta$ and the corresponding edge probabilities $\mathcal{P}_{i,j,m}(\gamma)$ for all node pairs and layers requires $O(Rn^2 M)$ operations per full pass, similar to standard CP factorization. Evaluating the score function in (3) has the same order, since it aggregates contributions over all $(i,j,m)$.

In most real-world settings, multilayer networks are sparse. Using sparse tensor representations and mini-batching over observed edges, the dominant cost per iteration scales as $O(R|E|)$, where $|E|$ is the total number of observed edges across all layers. The update of the working correlation $W$ in (8) is computed from aggregated residuals and can be performed infrequently (e.g., every $K$ epochs), so its amortized overhead is small compared to the main embedding updates. These properties make T-GINEE applicable to moderate-scale multilayer graphs (e.g., $n$ in the thousands and $M$ in the tens) when combined with sparse tensor implementations. Scaling to large-scale graphs necessitates efficient sampling strategies; a detailed discussion on this adaptation is provided in **Appendix** C.

## 3. Theoretical Results of T-GINEE

In this section, we present the foundational theoretical properties of the full-batch T-GINEE estimating equation. The mini-batch negative-sampling implementation used for massive graphs (Appendix C) is a practical scalable approximation evaluated empirically; extending the asymptotic guarantees to that regime remains an open problem. Full proofs are provided in **Appendix** D.

### 3.1. Assumptions

**Assumption 3.1** (Boundedness). There exist constants $0 < \varepsilon < 1/2$ and $C < \infty$ such that for all $(i,j,m)$ and all parameters $\gamma$ in a neighborhood of $\gamma_0$,

$$|\mathcal{A}_{i,j,m}| \le C, \qquad \varepsilon \le \mathcal{P}_{i,j,m}(\gamma) \le 1 - \varepsilon, \qquad |g'(\mathcal{P}_{i,j,m}(\gamma))| \le C \tag{9}$$

The derivatives of $g$ up to the required order are uniformly bounded near $\gamma_0$. For the sparsity-aware logit links in Section 2.3, the upper bound $1 - \varepsilon$ can be replaced by $s - \varepsilon$, where $s \in (0,1)$ is the sparsity coefficient.

This assumption keeps relevant random variables and their derivatives within controlled limits, preventing extreme values that could destabilize estimation near $\gamma_0$.

**Assumption 3.2** (Identifiability and Effective Sample Size). The true parameter tensor $\Theta_0$ admits a rank-$R$ CP decomposition that is identifiable up to permutation and scaling. Specifically, if the Kruskal ranks of $\alpha_0 \in \mathbb{R}^{n \times R}$ and $\beta_0 \in \mathbb{R}^{M \times R}$ satisfy $2\,k_\alpha + k_\beta \ge 2R + 2$, then the rank-$R$ CP decomposition of $\Theta_0$ is unique, justifying $\gamma_0 = [\mathrm{vec}(\alpha_0)^{\top}, \mathrm{vec}(\beta_0)^{\top}]^{\top}$ as a well-defined target.

Let $p_N = (n + M)R$ be the effective parameter dimension and $|E_{\mathrm{obs}}|$ the number of observed (non-zero) edge entries. We assume the Fisher information and relevant Hessians are well-conditioned, and that

$$\frac{p_N^2}{|E_{\mathrm{obs}}|} \longrightarrow 0 \qquad \text{as } n, M \to \infty. \tag{10}$$

This data-adaptive condition controls the third-order Taylor remainder in the proof of asymptotic normality. In the

dense regime $|E_{\text{obs}}| \asymp n^2$, it reduces to $p_N = o(n)$. For sparse graphs, $|E_{\text{obs}}|$—rather than $n$ alone—represents the effective sample size. This condition is restrictive and is not asserted to hold for the scalable mini-batch implementation.

We refer to the ratio $|E_{\text{obs}}|/p_N$ as the *overdetermination ratio*, a directly computable diagnostic used in our empirical study (Appendix C). Condition (10) should be interpreted as a restricted asymptotic regime sufficient for the full-batch expansion below.

**Assumption 3.3** (Working Covariance). The true covariance matrices $\Sigma_{i,j}$ are positive definite with eigenvalues bounded away from zero and infinity. The working covariance $\widehat{\Sigma}_{i,j}$ satisfies

$$\left\| \widehat{\Sigma}_{i,j}^{-1} - \widetilde{\Sigma}_{i,j}^{-1} \right\|_F = O_p(N^{-1/2})$$

for some positive definite $\widetilde{\Sigma}_{i,j}$ with bounded eigenvalues. Correlation misspecification is allowed at this rate.

Under the independent-pair assumption, $\widehat{W}$ is the average of $N$ approximately i.i.d. matrix-valued terms with finite second moments and thus converges at the parametric rate $O_p(N^{-1/2})$ (Van der Vaart, 2000).

**Assumption 3.4** (Moment Conditions). The standardized residuals have sub-Gaussian tails, and there exists $\delta > 0$ such that

$$\max_{i,j} \mathbb{E}\left[ \left\| \Sigma_{i,j}^{-1/2} \left( \mathcal{A}_{i,j,\cdot} - \mathcal{P}_{i,j,\cdot}(\Theta_0) \right) \right\|^{2+\delta} \right] < \infty.$$

This ensures Lindeberg-type (Ash & Doléans-Dade, 2000; Van der Vaart, 2000; Brown, 1971) conditions for central limit arguments.

**Assumption 3.5** (Smoothness). $\Theta_0 \in \mathbb{R}^{n \times n \times M}$ admits a unique rank-$R$ CP decomposition $\Theta_0 = \mathcal{I} \times_1 \alpha_0 \times_2 \alpha_0 \times_3 \beta_0$. The link $g$ is three-times continuously differentiable with uniformly bounded derivatives. The partial derivatives of $\mathcal{P}(\gamma)$ are bounded, and the Hessians with respect to $(\alpha, \beta)$ are well-conditioned near $\gamma_0$.

These smoothness conditions enable Taylor-expansion arguments and ensure the stability of the parameter estimates used in the proofs of consistency and asymptotic normality.

### 3.2. Main Theoretical Results

Define the score function

$$s(\gamma) = \sum_{i \leq j} \left( \frac{\partial \mathcal{P}_{i,j,\cdot}}{\partial \gamma} \right)^{\top} \widehat{\Sigma}_{i,j}^{-1} \left( \mathcal{A}_{i,j,\cdot} - \mathcal{P}_{i,j,\cdot}(\gamma) \right). \quad (11)$$

We now state consistency and asymptotic normality of the full-batch estimator. Consistency requires only Assumptions 3.1, 3.3–3.5 and the identifiability part of Assumption 3.2; asymptotic normality additionally invokes condition (10).

**Theorem 3.1** (Consistency). *Under Assumptions 3.1–3.5, there exists a solution $\hat{\gamma}$ to $s(\gamma) = 0$ with*

$$\|\hat{\gamma} - \gamma_0\|_2 = O_p(N^{-1/2}),$$

*where $N = n(n+1)/2$. Consistency does not rely on condition* (10).

*Proof.* See Appendix D.2. $\square$

**Theorem 3.2** (Asymptotic normality). *Under Assumptions 3.1–3.5, including $p_N^2/|E_{\text{obs}}| \to 0$,*

$$\sqrt{N}\,(\hat{\gamma} - \gamma_0) \xrightarrow{d} \mathcal{N}(0, \Omega),$$

*where $\Omega = M(\gamma_0)^{-1} B(\gamma_0) M(\gamma_0)^{-1}$, with $M(\gamma_0)$ the limit of $N^{-1}\nabla s(\gamma_0)$ and $B(\gamma_0)$ the limit of $\operatorname{Var}(N^{-1/2}s(\gamma_0))$. In the dense regime $|E_{\text{obs}}| \asymp N \asymp n^2$, the condition reduces to $p_N = o(n)$; in sparse regimes it should be evaluated through the overdetermination ratio $|E_{\text{obs}}|/p_N$.*

*Proof.* See Appendix D.3. $\square$

**Corollary 3.3.** *Under Assumptions 3.1–3.5, if $\widetilde{s}(\gamma)$ uses $\widetilde{\Sigma}_{i,j}^{-1}$ in place of $\widehat{\Sigma}_{i,j}^{-1}$, then*

$$\|s(\gamma_0) - \widetilde{s}(\gamma_0)\| = O_p(\sqrt{N}).$$

*Proof.* See Appendix D.4. $\square$

Theorem 3.1 ensures $\hat{\gamma} \to \gamma_0$ at rate $N^{-1/2}$ (equivalently $1/n$) without invoking (10), so it applies broadly across the configurations in Section 4. Theorem 3.2 characterizes the limiting distribution under the data-adaptive growth condition; the overdetermination ratio $|E_{\text{obs}}|/p_N$ serves as a computable diagnostic for whether normality is expected to be reliable. In our empirical study (Appendix C), this ratio predicts estimation stability across six orders of magnitude in graph size: higher ratios yield smaller AUC standard deviation, while values near or below 1 show larger variance, consistent with the theory. Finally, Corollary 3.3 shows that estimating $\Sigma_{i,j}^{-1}$ at rate $O_p(N^{-1/2})$ yields only an $O_p(\sqrt{N})$ perturbation of the score at $\gamma_0$, justifying the momentum-based estimation of $W$ in lieu of an oracle covariance. Due to space limitations, additional remarks and discussions are provided in **Appendix** E.

## 4. Experiments

In this section, we conduct comprehensive experiments to evaluate our T-GINEE framework.

*Table 1.* Link prediction performance (AUC) on synthetic multilayer network.

| Method | CP | Tucker | NMF | SVD | LSE | MASE | NNTUCK | SPECK | HOSVD | T-GINEE |
|--------|------|--------|------|------|------|------|--------|-------|-------|---------|
| AUC | 0.4488 | 0.5291 | 0.7216 | 0.8130 | 0.2234 | 0.3821 | 0.6105 | 0.7603 | 0.8503 | **0.9395** |

## 4.1. Experiment Settings

To comprehensively evaluate the performance of our proposed **T-GINEE** model, we conduct experiments on four benchmark multilayer network datasets, each capturing distinct real-world relational structures. Due to space limitations, detailed descriptions of datasets, baselines, and implementation details are provided in **Appendix F**.

## 4.2. Synthetic Data Results

To evaluate our method in a controlled environment, we generate synthetic multilayer networks with known correlation structures using a parameterized model:

$$\mathcal{P}_{i,j,m} = \rho \cdot \mathcal{P}_{i,j}^{\text{base}} + (1-\rho) \cdot U_{i,j,m}, \qquad \mathcal{A}_{i,j,m} = \mathbf{1}\{V_{i,j,m} < \mathcal{P}_{i,j,m}\}, \tag{12}$$

Here, $\rho = 0.2$ controls inter-layer correlation, $\mathcal{P}_{i,j}^{\text{base}}$ is a shared base probability matrix, and $U_{i,j,m}$ and $V_{i,j,m}$ are i.i.d. $\text{Unif}[0,1]$ noise variables. We construct networks with $n = 100$ nodes and $M = 3$ layers for link prediction tasks.

As shown in Table 1, T-GINEE achieves the highest AUC score of 0.9395, substantially outperforming all baselines including the second-best HOSVD. The significant performance gap between tensor-based methods and simpler approaches like LSE (0.2234) and MASE (0.3821) confirms the importance of explicitly modeling multilayer dependencies. Furthermore, the dramatic improvement of T-GINEE over basic CP decomposition (0.4488) demonstrates the effectiveness of our statistical regularization framework in capturing complex inter-layer correlations.

We also evaluate a heterogeneous synthetic setting with layer-dependent correlations and block-structured base probabilities. This experiment is designed to test whether the learned working covariance can recover nonuniform inter-layer structure rather than only homogeneous correlations. At $N/((n + M)R) \approx 1.5$, T-GINEE obtains AUC 0.5822, while the CP baseline obtains 0.5897, a difference within one standard deviation in this low-overdetermination regime. More importantly, the learned $W$ recovers the correct ordering of inter-layer similarities: the L0–L2 pair receives the highest off-diagonal weight ($W = 0.218$, Jaccard $= 0.097$), while L0–L1 receives a lower weight ($W = 0.206$, Jaccard $= 0.035$). This further confirms that the covariance component effectively captures meaningful inter-layer structure, even when the observed AUC gains are relatively modest.

*Table 2.* Performance comparison of different methods. "oom" denotes Out-Of-Memory errors. Bold indicates the best result in each column and underline indicates the second-best result.

| Method | AUC score on different datasets | | | | | |
|--------|------|-----------|------|-------|--------|---------------|
| | AUCS | Krackhardt | WAT | Yeast | dblp | stackoverflow |
| CP | 0.374 | 0.354 | 0.454 | 0.397 | oom | oom |
| Tucker | 0.487 | 0.702 | 0.580 | 0.745 | oom | oom |
| NMF | 0.848 | 0.921 | 0.707 | 0.863 | **0.6505** | 0.9642 |
| SVD | 0.877 | 0.932 | 0.719 | 0.879 | 0.6093 | 0.9682 |
| LSE | 0.297 | 0.384 | 0.153 | 0.047 | 0.6302 | oom |
| MASE | 0.480 | 0.361 | 0.342 | 0.347 | oom | oom |
| NNTUCK | 0.500 | 0.521 | 0.741 | 0.667 | oom | oom |
| SPECK | 0.793 | 0.658 | 0.655 | 0.903 | oom | oom |
| HOSVD | 0.897 | 0.783 | 0.820 | 0.902 | oom | oom |
| T-GINEE | **0.920** | **0.948** | **0.838** | **0.921** | 0.6478 | **0.9831** |

## 4.3. Real-World Results

Based on the experimental results shown in Table 2, T-GINEE demonstrates strong performance across datasets of varying scales and complexities compared to baseline methods. On the standard benchmark datasets (AUCS, Krackhardt, WAT, and Yeast), T-GINEE achieves the highest AUC scores of 0.920, 0.948, 0.838, and 0.921 respectively, outperforming all baseline methods in our experiments. Among the baselines, HOSVD is second best on AUCS (0.897) and WAT (0.820), SVD is second best on Krackhardt (0.932), and SPECK is second best on Yeast (0.903). Traditional matrix factorization approaches such as SVD and NMF also show competitive performance, whereas simpler methods such as CP decomposition and LSE exhibit limited effectiveness, with LSE performing poorly on Yeast.

To comprehensively evaluate the scalability of T-GINEE, we extended our experiments to two large-scale real-world networks: DBLP, an academic collaboration network with up to 300,000 nodes, and Stack Overflow, a massive temporal interaction network with approximately 2.6 million nodes. As indicated in the rightmost columns of Table 2, most traditional tensor-based methods, including CP, Tucker, HOSVD, and MASE, failed to process these large-scale datasets, resulting in Out-Of-Memory (OOM) errors. This highlights the severe computational bottleneck of standard tensor decompositions when applied to million-node graphs. In contrast, T-GINEE successfully scaled to these massive datasets. On DBLP, T-GINEE achieved an AUC of 0.6478, slightly below NMF (0.6505) but above SVD (0.6093), LSE (0.6302), and the tensor baselines that ran out of memory. On Stack Overflow, T-GINEE achieved the best AUC (0.9831), exceeding NMF (0.9642) and SVD (0.9682).

The performance gains on the standard benchmarks and Stack Overflow can be attributed to T-GINEE's ability to

*Table 3.* Comparison with GNN baselines on link prediction (AUC) and community detection (NMI). Bold indicates the best result in each row.

| Dataset | Task | T-GINEE | GCN | GraphSAGE | MGCN | MR-GCN |
|---------|------|---------|-----|-----------|------|--------|
| Karate | AUC | 0.817 | **0.834** | 0.831 | 0.824 | 0.809 |
| Karate | NMI | **0.837** | 0.206 | 0.183 | 0.206 | 0.732 |
| Dolphins | AUC | 0.869 | **0.872** | 0.801 | 0.625 | 0.604 |
| Lesmis | AUC | 0.786 | **0.794** | 0.770 | 0.740 | 0.728 |
| Polbooks | AUC | **0.973** | 0.887 | 0.841 | 0.887 | 0.923 |
| Polbooks | NMI | **0.428** | 0.021 | 0.018 | 0.021 | 0.176 |
| Email | AUC | **0.958** | 0.873 | 0.832 | 0.873 | 0.846 |
| Email | NMI | **0.641** | 0.429 | 0.391 | 0.429 | 0.101 |

*Table 4.* New-layer generalization on DBLP-5K. T-GINEE predicts the fifth layer with zero observed training edges from that layer.

| Model | Layer-5 edges used | AUC |
|-------|--------------------|-----|
| T-GINEE zero-shot | 0 | **0.7733 ± 0.0194** |
| MGCN retrained | 8,297 | 0.7089 ± 0.0200 |
| MR-GCN retrained | 8,297 | 0.6306 ± 0.0155 |

capture high-order structural patterns and incorporate cross-layer residual dependence through the working covariance. The DBLP result is intentionally reported without over-claiming: a simple NMF baseline is marginally higher in AUC, while T-GINEE remains competitive and provides a covariance-aware tensor representation. Notably, the performance improvement is particularly pronounced on the WAT dataset, where T-GINEE outperforms the second-best method by a margin of $0.018$ in AUC, suggesting that our model is especially effective in handling complex network structures. For further evidence of T-GINEE's effectiveness, see the triangle prediction analysis in **Appendix** G.

**GNN baselines and community detection.** Table 3 compares T-GINEE with single-layer GNN baselines (GCN and GraphSAGE on aggregated graphs) and multiplex GNN baselines (MGCN and MR-GCN) on link prediction and community detection. GCN is competitive on very small graphs, but T-GINEE is strongest on larger benchmarks and achieves the best NMI on all labeled datasets. No single GNN baseline dominates T-GINEE across both tasks.

**New-layer generalization.** We further evaluate zero-shot transfer to an unseen layer on DBLP-5K. T-GINEE is trained on four layers and asked to predict links on the fifth layer with zero observed edges from that layer. As shown in Table 4, T-GINEE outperforms retrained MGCN and MR-GCN even though those baselines observe all layer-5 edges during retraining. This behavior follows from the scoring function $\langle \alpha_i \odot \alpha_j, \beta_m \rangle$, which conditions on the layer index $m$ while sharing node embeddings across layers.

*Table 5.* Stability diagnostics for sparsity and large-scale settings.

| Setting | Density | $N/((n+M)R)$ | AUC std | Interpretation |
|---------|---------|--------------|---------|----------------|
| Synthetic ($1\%$, $n=100$) | $1\%$ | 0.015 | 0.0076 | underdetermined |
| Synthetic ($3\%$, $n=500$) | $3\%$ | 0.73 | 0.0781 | ill-conditioned |
| Synthetic ($5\%$, $n=500$) | $5\%$ | 1.22 | 0.0421 | near-determined |
| Synthetic ($15\%$, $n=500$) | $15\%$ | 3.67 | 0.0312 | overdetermined |
| DBLP | $0.0011\%$ | 0.215 | 0.0105 | moderate |
| Stack Overflow | $0.1\%$ | 1.16 | 0.0011 | near-determined |

### 4.4. Sensitivity to Sparsity

We further study how T-GINEE performs as multilayer networks vary in sparsity. Using the synthetic setup in Section 4.2, we adjust the Bernoulli generator so the proportion of observed edges (i.e., $1$ entries in the adjacency tensor) ranges from below $1\%$ to about $15\%$. At each sparsity level, we train T-GINEE with the hyperparameters from Section 4.2 and run 5 trials with different seeds. Figure 2 reports mean test AUC with one standard deviation. AUC increases from the extremely sparse regime to moderate sparsity, peaking at roughly AUC $\approx 0.75$, then remains fairly stable and declines only gradually for dense graphs, staying above AUC $\approx 0.63$ even at $10$–$15\%$ edge density. Overall, T-GINEE is robust across a broad sparsity range: performance is weaker and more variable when edges are extremely sparse, but stabilizes once a moderate amount of edge information is available.

**Overdetermination and stability.** The sparsity experiment also clarifies why absolute edge count matters in addition to density. The key quantity is the overdetermination ratio $N/((n+M)R)$: when it is well above 1, estimation is stable; when it approaches or falls below 1, the system becomes more ill-conditioned. In additional synthetic experiments with $n=500$ and three layers, AUC standard deviation decreases from 0.0781 at density $3\%$ and ratio 0.73 to 0.0312 at density $15\%$ and ratio 3.67. On real large-scale graphs, DBLP is much sparser than the synthetic $1\%$ setting but contains far more absolute edges, achieving AUC std 0.0105; Stack Overflow achieves AUC std 0.0011 despite high sparsity. These results indicate that stability is governed by the amount of structural information relative to parameter dimension, not by edge density alone.

Due to space constraints, a detailed investigation into the impact of embedding dimension and regularization weight on model performance and computational efficiency is presented in **Appendix** I.

## 5. Related Work

**Network embedding.** Network embedding learns low-dimensional node representations that preserve network structure. Early methods include matrix factorization techniques such as SVD (Golub & Reinsch, 1971) and NMF (Lee & Seung, 2000; Cai et al., 2010). Random

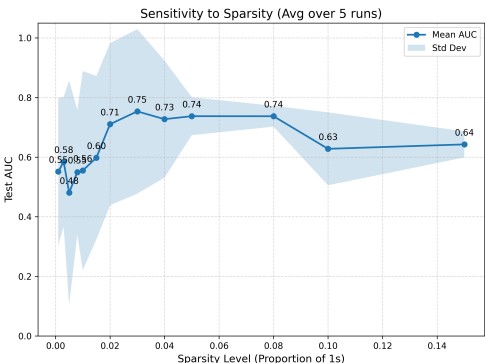

*Figure 2.* Sensitivity of T-GINEE to graph sparsity on synthetic multilayer networks. We vary the proportion of observed edges (proportion of 1 entries) and report the mean test AUC over 5 runs with one standard deviation as the shaded region.

walk-based approaches, including DeepWalk (Perozzi et al., 2014) and node2vec (Grover & Leskovec, 2016), adapt word embedding techniques to networks. More recently, graph convolutional networks (GCNs) (Hamilton et al., 2017) have become popular for incorporating node features and modeling complex relations. However, these methods are ill-suited for multilayer systems because they either treat layers independently or use simplistic aggregation, losing inter-layer dependencies (Wang et al., 2017b; Dong et al., 2017). Such aggregation, often simple summation or concatenation of layer-specific embeddings, can obscure the distinct and complementary roles of different relationship types. T-GINEE leverages generalized estimating equations to explicitly capture cross-layer dependencies.

**Multilayer graph analysis and embedding.** Multilayer graphs provide rich representations for complex systems (Kivelä et al., 2014; De Domenico et al., 2013; Boccaletti et al., 2014), with tensor methods such as CP decomposition serving as natural extensions (Wang et al., 2017a). A recent survey (Yousefzadeh et al., 2025) highlights three key limitations of existing embedding methods: (i) ignoring cross-layer dependencies (Papalexakis et al., 2013; Boden et al., 2017); (ii) assuming complete node correspondence; and (iii) lacking theoretical guarantees in deep models (Song & Thiagarajan, 2018; Liu et al., 2017; Ghorbani et al., 2019; Huang et al., 2020). T-GINEE addresses (i) and (iii) by combining CP decomposition with generalized estimating equations (GEE)(Liang & Zeger, 1986), though it still assumes a shared node set across layers. Unlike prior matrix factorization(Tang et al., 2009; Gligorijević et al., 2016), random walk (Song & Thiagarajan, 2018; Liu et al., 2017), or GCN-based methods (Ghorbani et al., 2019; Huang et al., 2020), our framework explicitly models inter-layer correlations with rigorous guarantees of consistency and asymptotic normality under mild conditions, offering a statistically grounded complement to deep encoders.

# 6. Conclusion

We propose T-GINEE, a tensor-based generalized estimating equation framework for multilayer graph representation learning that explicitly models cross-network dependencies through a principled statistical formulation. By combining CP decomposition with GEE, T-GINEE establishes consistency and asymptotic normality for the full-batch estimating-equation formulation under the stated assumptions, while the scalable mini-batch implementation is evaluated empirically as a practical extension. This bridges classical inference and modern representation learning, offering interpretability and scalability. Experiments demonstrate its effectiveness on synthetic and real-world networks, including additional GNN baselines, community detection, covariance ablations, noise-robustness tests, and large-scale profiling. Limitations include the restricted scope of the current asymptotic theory, reliance on shared node alignment, and potential instability in highly underdetermined sparse regimes. Future work will explore formal theory for mini-batch negative sampling, integration with deep encoders, and extensions to partially aligned or heterogeneous multilayer networks. Details about limitations and LLM usage are provided in **Appendix** J and K.

# Impact Statement

This paper presents work whose goal is to advance the field of machine learning. There are many potential societal consequences of our work, none of which we feel must be specifically highlighted here.

# Acknowledgements

The authors would like to express their sincere gratitude to Prof. Junhui Wang (Department of Statistics, The Chinese University of Hong Kong) and Prof. Yaoming Zhen (School of Data Science, The Chinese University of Hong Kong, Shenzhen) for their many insightful discussions and valuable suggestions. This research was partially supported by the National Natural Science Foundation of China (No. 62502404); the Hong Kong Research Grants Council under the Research Impact Fund (No. R1015-23), the Collaborative Research Fund (No. C1043-24GF), and the General Research Fund (No. 11218325); the Institute of Digital Medicine of City University of Hong Kong (No. 9229503); the Huawei Innovation Research Program; the Tencent Rhino-Bird Focused Research Program and the Tencent University Cooperation Project; the CCF-Kuaishou Large Model Explorer Fund (No. 2025008) and the Kuaishou University Cooperation Project; the CCF-Didi Gaia Scholars Research Fund; and ByteDance. The authors gratefully acknowledge these organizations for their generous support, which made this work possible.

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

# A. Proof of Derivations $\frac{\partial \operatorname{vec}(\Theta)}{\partial \gamma}$

We first consider a rank-one tensor $\mathcal{T} = a \circ a \circ c$ with $a \in \mathbb{R}^n$ and $c \in \mathbb{R}^M$. By the definition of $\operatorname{vec}(\mathcal{T})$, we have

$$\operatorname{vec}(\mathcal{T}) = \left(c_1(a \otimes a)^\top, \ldots, c_M(a \otimes a)^\top\right)^\top.$$

Denote $\{e_i\}_{i=1}^n$ as the canonical basis in $\mathbb{R}^n$. For one thing, note that

$$\frac{\partial(a \otimes a)}{\partial a} = \begin{bmatrix} a^\top + a_1 e_1^\top & a_2 e_1^\top & \cdots & a_n e_1^\top \\ a_1 e_2^\top & a^\top + a_2 e_2^\top & \cdots & a_n e_2^\top \\ \vdots & \vdots & \ddots & \vdots \\ a_1 e_n^\top & a_2 e_n^\top & \cdots & a^\top + a_n e_n^\top \end{bmatrix}^\top$$

$$= I_n \otimes a + a \otimes I_n, \tag{13}$$

which leads to

$$\frac{\partial \operatorname{vec}(\mathcal{T})}{\partial a} = c^\top \otimes \left(I_n \otimes a + a \otimes I_n\right). \tag{14}$$

For another, it is clear that

$$\frac{\partial \operatorname{vec}(\mathcal{T})}{\partial c} = I_M \otimes (a \otimes a). \tag{15}$$

According to the CP decomposition of $\Theta$, we have

$$\operatorname{vec}(\Theta) = \sum_{r=1}^R \left(\beta_1^{(r)}(\alpha^{(r)} \otimes \alpha^{(r)})^\top, \ldots, \beta_M^{(r)}(\alpha^{(r)} \otimes \alpha^{(r)})^\top\right)^\top.$$

By the property (14), we have

$$\frac{\partial \operatorname{vec}(\Theta)}{\partial \operatorname{vec}(\alpha)} = \begin{bmatrix} (\beta^{(1)})^\top \otimes \left(I_n \otimes \alpha^{(1)} + \alpha^{(1)} \otimes I_n\right) \\ (\beta^{(2)})^\top \otimes \left(I_n \otimes \alpha^{(2)} + \alpha^{(2)} \otimes I_n\right) \\ \vdots \\ (\beta^{(R)})^\top \otimes \left(I_n \otimes \alpha^{(R)} + \alpha^{(R)} \otimes I_n\right) \end{bmatrix}. \tag{16}$$

By the property (15), we have

$$\frac{\partial \operatorname{vec}(\Theta)}{\partial \operatorname{vec}(\beta)} = \begin{bmatrix} I_M \otimes \left(\alpha^{(1)} \otimes \alpha^{(1)}\right) \\ I_M \otimes \left(\alpha^{(2)} \otimes \alpha^{(2)}\right) \\ \vdots \\ I_M \otimes \left(\alpha^{(R)} \otimes \alpha^{(R)}\right) \end{bmatrix} = I_M \otimes \left(\alpha \odot_{\mathrm{KR}} \alpha\right), \tag{17}$$

The desired result follows from (16) and (17) immediately.

# B. Full Formulation and Approximation of T-GINEE

Putting all pieces together, the T-GINEE in (3) is approximated by

$$\sum_{i \leq j} \begin{bmatrix} (\beta^{(1)})^\top \otimes \Delta^{(1)} \\ \vdots \\ (\beta^{(R)})^\top \otimes \Delta^{(R)} \\ I_M \otimes (\alpha \odot_{\mathrm{KR}} \alpha) \end{bmatrix} \begin{bmatrix} \operatorname{vec}\left(\mathcal{E}^{(i,j,1)}\right)^\top \\ \vdots \\ \operatorname{vec}\left(\mathcal{E}^{(i,j,M)}\right)^\top \end{bmatrix}^\top \times \operatorname{diag}\left(g'(\mathcal{P}_{i,j,1}), \ldots, g'(\mathcal{P}_{i,j,M})\right)^{-1} \widehat{\Sigma}_{i,j}^{-1} \left(\mathcal{A}_{i,j,\cdot} - \mathcal{P}_{i,j,\cdot}(\gamma)\right) = \mathbf{0},$$

where $\Delta^{(r)} = I_n \otimes (\alpha^{(r)})^\top + (\alpha^{(r)})^\top \otimes I_n$ for $r \in [R]$ and $\widehat{\Sigma}_{i,j} = \Gamma_{i,j}^{1/2} \widehat{W} \Gamma_{i,j}^{1/2}$ is the estimated covariance matrix.

**Kruskal-type uniqueness.** The CP decomposition $\Theta_0 = \mathcal{I} \times_1 \alpha_0 \times_2 \alpha_0 \times_3 \beta_0$ used throughout this paper is identifiable up to permutation and scaling under the symmetric Kruskal-type condition $2\,k_{\alpha_0} + k_{\beta_0} \geq 2R + 2$, where $k_{\alpha_0}$ and $k_{\beta_0}$ denote the Kruskal ranks of $\alpha_0 \in \mathbb{R}^{n \times R}$ and $\beta_0 \in \mathbb{R}^{M \times R}$, respectively (see, e.g., (Kolda & Bader, 2009)). This is the precise sufficient condition referenced in Assumption 3.2.

## C. Scalability to Massive Graphs

Scaling tensor-based methods to massive graphs (e.g., $n > 10^6$ nodes) presents significant challenges. Standard tensor decompositions typically operate on the full adjacency tensor $\mathcal{A} \in \mathbb{R}^{n \times n \times M}$, where $n$ is the number of nodes. For massive datasets like Stack Overflow ($n \approx 2.6 \times 10^6$), instantiating this tensor is infeasible. Instead, we adopt an edge-centric perspective. Let $\mathcal{E}^+$ denote the set of observed positive edges across all layers. We define a sparse dataset $\mathcal{D} = \{(u, v, m) \mid \mathcal{A}_{u,v,m} = 1\}$.

**Dynamic Negative Sampling.** To avoid the computationally prohibitive cost of iterating over all non-existent edges (zeros in the tensor), we employ dynamic negative sampling during training only for the largest sparse graphs in our experiments (DBLP and Stack Overflow). For the small- and medium-scale datasets (AUCS, Krackhardt, WAT, and Yeast), we enumerate node pairs directly and do not use negative sampling. For each mini-batch of positive samples $\mathcal{B}^+ \subset \mathcal{E}^+$, we generate a corresponding set of negative samples $\mathcal{B}^-$. For a positive triplet $(u, v, m) \in \mathcal{B}^+$, we sample a negative node $v' \in \mathcal{V}$ uniformly at random to construct $(u, v', m)$. The assumption is not that every sampled non-edge is guaranteed to be negative; rather, under uniform sampling in a sufficiently sparse layer, the false-negative probability is approximately the edge density $\rho$, so collisions are rare when $\rho \ll 1$. This strategy reduces the per-epoch computational complexity from $O(n^2 \cdot M)$ to $O(|\mathcal{E}| \cdot M)$ in the sparse large-scale regime. The resulting mini-batch estimator is a scalable practical approximation to the full-batch GEE objective and is not covered by the current full-batch asymptotic theory.

### C.1. Batch-wise GEE with Momentum Update

The core of T-GINEE is the Generalized Estimating Equation (GEE) framework, which traditionally requires estimating the working covariance matrix $\mathbf{W}$ using residuals from all node pairs. To adapt this to mini-batch training, we propose a **Batch-GEE Loss** combined with a momentum update mechanism.

**Batch-GEE Loss Function.** We reformulate the global GEE objective into a differentiable loss function calculated over a mini-batch $\mathcal{B} = \mathcal{B}^+ \cup \mathcal{B}^-$. The total loss $\mathcal{L}_{total}$ is defined as:

$$\mathcal{L}_{total} = \mathcal{L}_{BCE} + \lambda \cdot \mathcal{L}_{GEE}(\mathcal{B}; \mathbf{W}), \tag{18}$$

where $\mathcal{L}_{BCE}$ is the standard binary cross-entropy loss measuring reconstruction error. The regularization term $\mathcal{L}_{GEE}$ captures inter-layer correlations and is defined as:

$$\mathcal{L}_{GEE}(\mathcal{B}; \mathbf{W}) = \frac{1}{|\mathcal{B}|} \sum_{(i,j) \in \mathcal{B}} \mathbf{r}_{ij}^T \mathbf{\Sigma}_{ij}^{-1} \mathbf{r}_{ij}, \tag{19}$$

where $\mathbf{r}_{ij} \in \mathbb{R}^M$ is the residual vector for node pair $(i, j)$ across $M$ layers, calculated as $\mathbf{r}_{ij} = \mathbf{a}_{ij} - \mathbf{p}_{ij}(\gamma)$. Here, $\mathbf{a}_{ij}$ is the observed edge vector and $\mathbf{p}_{ij}$ is the predicted probability vector. The matrix $\mathbf{\Sigma}_{ij}^{-1}$ is the inverse of the working covariance for pair $(i, j)$, approximated via the global correlation structure $\mathbf{W}$. Specifically, we utilize the standardized residuals $\tilde{\mathbf{r}}_{ij} = \mathbf{\Gamma}_{ij}^{-1/2} \mathbf{r}_{ij}$, where $\mathbf{\Gamma}_{ij}$ is the diagonal variance matrix derived from the predicted probabilities. The GEE term simplifies to the quadratic form $\tilde{\mathbf{r}}_{ij}^T \mathbf{W}^{-1} \tilde{\mathbf{r}}_{ij}$.

**Momentum Update for Working Covariance.** A critical challenge in batch training is that the global correlation structure $\mathbf{W}$ cannot be accurately estimated from a single batch. To resolve this, we maintain a global buffer for $\mathbf{W}$ and update it using a momentum-based moving average. In iteration $t$, let $\mathbf{W}_{batch}^{(t)}$ be the empirical correlation matrix computed from the current batch's standardized residuals. The global $\mathbf{W}$ is updated as:

$$\mathbf{W}^{(t)} \leftarrow \alpha \mathbf{W}^{(t-1)} + (1 - \alpha) \mathbf{W}_{batch}^{(t)}, \tag{20}$$

where $\alpha \in [0, 1)$ is the momentum coefficient (set to 0.9 in our experiments). This approach ensures that $\mathbf{W}$ stabilizes to represent the global cross-layer dependency structure while allowing efficient mini-batch optimization. The matrix $\mathbf{W}$

*Table 6.* Large-scale profiling of the scalable T-GINEE implementation. Runtime is reported per epoch.

| Dataset | Nodes | Edges | AUC | AUC std | Time/epoch | Peak memory |
|---|---|---|---|---|---|---|
| DBLP | 300K | 1,032,786 | 0.6244 | 0.0105 | 8.3s | 54 MB |
| Stack Overflow | 2.58M | 47,903,266 | 0.9538 | 0.0011 | 753s | 795 MB |

*Table 7.* Effect of negative sampling ratio on DBLP over five seeds.

| Negative sampling ratio | AUC mean | AUC std |
|---|---|---|
| 1:1 | 0.6027 | 0.0526 |
| 1:3 | **0.6659** | **0.0313** |
| 1:5 | 0.6468 | 0.0592 |
| 1:10 | 0.6221 | 0.0490 |

is only $M \times M$ and is maintained as a single layer-level working correlation matrix; pair-specific covariances $\widehat{\Sigma}_{i,j}$ are formed on the fly from $\mathbf{W}$ and the predicted probabilities. Thus, the scalable implementation does not instantiate any $n \times n$ covariance object.

### C.2. Complexity Analysis

The Scalable T-GINEE framework significantly reduces resource requirements. Regarding **Space Complexity**, by storing only the model parameters (embeddings $\alpha \in \mathbb{R}^{n \times R}$, $\beta \in \mathbb{R}^{M \times R}$) and the edge list, the space complexity is $O((n + M)R + |\mathcal{D}|)$, which is linear with respect to the number of nodes and observed edges, avoiding the $O(n^2)$ bottleneck. As for **Time Complexity**, the cost per training iteration is proportional to the batch size $|\mathcal{B}|$. The total time complexity per epoch is $O(|\mathcal{D}| \cdot R)$, i.e., linear in the number of observed edge triplets and the embedding rank, making it feasible to train on datasets with millions of nodes (e.g., Stack Overflow) using standard GPU or even CPU hardware.

### C.3. Large-scale Profiling and Sampling Ablation

Table 6 reports empirical profiling on the large-scale datasets. On Stack Overflow, T-GINEE trains on approximately 2.58 million nodes and 47.9 million observed edge triplets with less than 1GB peak memory. The model parameters are dominated by the node embedding matrix, while the layer embedding matrix and the working covariance matrix are negligible because $M$ is small. Conventional dense tensor baselines such as Tucker, HOSVD, and NNTuck fail with out-of-memory errors on these datasets because they require materializing or repeatedly scanning dense tensor objects.

We also ablate the negative sampling ratio on DBLP using five independent seeds. As shown in Table 7, the 1:3 ratio gives the highest mean AUC and lowest standard deviation, balancing positive signal and negative contrast. Larger ratios produce diminishing returns and higher variance, consistent with gradient dilution from excessive negatives. Across these ratios, the learned working covariance remains stable and close to diagonal, indicating that covariance estimation is robust to reasonable choices of the sampling ratio.

## D. Derivations of Theorems

**Notation and effective sample size.** Throughout this appendix, $N = n(n + 1)/2$ denotes the number of node pairs $(i, j)$ with $i \leq j$, $p_N = (n + M)R$ denotes the effective parameter dimension, and $|E_{\text{obs}}|$ denotes the number of observed (non-zero) edge entries used in the full-batch estimating equation. We use the standard stochastic-order notation $O_p(\cdot)$ and $o_p(\cdot)$ throughout, and we adopt the *independent-pair* working assumption stated in Section 2.3: conditional on the parameters, the edge vectors $\{\mathcal{A}_{i,j,\cdot} : i \leq j\}$ are independent across node pairs $(i, j)$. This is a standard working assumption shared by stochastic block models and latent space models, and is used in the moment computations below. It does not require independence *within* a pair across layers $m \in [M]$; cross-layer dependence within a pair is exactly what the working covariance $\widehat{\Sigma}_{i,j}$ models.

## D.1. Lemmas

**Lemma D.1.** *Let $N = n(n+1)/2$ denote the number of node pairs $(i, j)$ with $i \leq j$. Under Assumptions 3.1–3.5, consider the initial estimator obtained by solving*

$$\sum_{i \leq j} \left( \frac{\partial \mathcal{P}_{i,j,\cdot}}{\partial \gamma} \right)^{\top} (\mathcal{A}_{i,j,\cdot} - \mathcal{P}_{i,j,\cdot}(\gamma)) = 0,$$

*using an independence working structure $(\Sigma_{i,j} = I_M)$. Then, the initial estimator $\tilde{\gamma}$ is $O_p(N^{-1/2})$-consistent for the true parameter $\gamma_0$.*

*Proof.* Consider the estimating equation defined by

$$s(\gamma) = \sum_{i \leq j} \left( \frac{\partial \mathcal{P}_{i,j,\cdot}}{\partial \gamma} \right)^{\top} (\mathcal{A}_{i,j,\cdot} - \mathcal{P}_{i,j,\cdot}(\gamma)).$$

Under Assumption 3.1, all random variables involved, including $\mathcal{A}_{i,j,m}$, $\mathcal{P}_{i,j,m}(\gamma)$, and the derivatives of the link function $g$, are uniformly bounded for all indices $(i, j, m)$ and for all parameter vectors $\gamma$ in a neighborhood of the true parameter $\gamma_0$. Assumption 3.5 ensures that the partial derivatives $\frac{\partial \mathcal{P}_{i,j,\cdot}(\gamma)}{\partial \gamma}$ are bounded and that the link function $g$ is thrice continuously differentiable with uniformly bounded first and second derivatives.

By the law of large numbers, as $n$ (and hence $N = n(n+1)/2$) tends to infinity, the normalized sum $N^{-1}s(\gamma)$ converges in probability to its expectation. Specifically, at the true parameter value $\gamma_0$, the expectation of each term in the sum satisfies

$$\mathbb{E}\left[ \left( \frac{\partial \mathcal{P}_{i,j,\cdot}}{\partial \gamma} \right)^{\top} (\mathcal{A}_{i,j,\cdot} - \mathcal{P}_{i,j,\cdot}(\gamma_0)) \right] = 0,$$

since $\mathbb{E}[\mathcal{A}_{i,j,\cdot}] = \mathcal{P}_{i,j,\cdot}(\gamma_0)$ by the model specification.

Assumption 3.2 guarantees that the true parameter $\gamma_0$ is uniquely identifiable, up to permutation and scaling of CP factors, as the solution to $\mathbb{E}\{s(\gamma)\} = 0$. Because consistency only requires the well-conditioning part of Assumption 3.2 together with the moment and smoothness assumptions, the data-adaptive growth condition $p_N^2/|E_{\text{obs}}| \to 0$ is *not* needed for this lemma. The identifiability and well-conditioning conditions ensure that the dimensionality does not impede the identification or the invertibility of the population Jacobian $M(\gamma_0)$.

Define the normalized Jacobian

$$D_N(\gamma) := -N^{-1} \frac{\partial s(\gamma)}{\partial \gamma}.$$

Expanding $N^{-1}s(\gamma)$ around $\gamma_0$ using a Taylor series, we obtain

$$N^{-1}s(\tilde{\gamma}) = N^{-1}s(\gamma_0) + D_N(\gamma_0)(\tilde{\gamma} - \gamma_0) + r_N,$$

where $\|r_N\| = o_p(N^{-1/2})$ by Assumption 3.5. From Lemma D.2, $D_N(\gamma_0)$ is invertible with eigenvalues bounded away from zero and infinity, ensuring that $D_N(\gamma_0)$ is well-conditioned.

Solving the linear approximation for $\tilde{\gamma}$ yields

$$\tilde{\gamma} - \gamma_0 = -D_N(\gamma_0)^{-1} N^{-1} s(\gamma_0) + o_p(N^{-1/2}).$$

The term $N^{-1/2}s(\gamma_0)$ is a sum of $N$ mean-zero random vectors with uniformly bounded moments; under Assumption 3.4 and by the central limit theorem,

$$N^{-1/2}s(\gamma_0) = O_p(1).$$

Since $D_N(\gamma_0)$ is non-singular and its inverse has bounded operator norm, it follows that

$$\tilde{\gamma} - \gamma_0 = O_p(N^{-1/2}).$$

Therefore, the initial estimator $\tilde{\gamma}$ converges to the true parameter $\gamma_0$ at the rate $N^{-1/2}$ in probability, establishing its $O_p(N^{-1/2})$-consistency. $\square$

**Lemma D.2.** *Let $N = n(n+1)/2$ and define the normalized Jacobian*

$$D_N(\gamma) = -N^{-1} \frac{\partial s(\gamma)}{\partial \gamma}.$$

*Under Assumptions 3.1–3.5, the matrix $D_N(\gamma_0)$ is invertible with eigenvalues bounded away from zero and infinity. Furthermore,*

$$\sup_{\|\gamma-\gamma_0\| \leq 4N^{-1/2}} \left\| D_N(\gamma) - D_N(\gamma_0) \right\| = O_p\big(N^{-1/2}\big).$$

*Proof.* By definition,

$$
\begin{aligned}
D_N(\gamma) &= -N^{-1} \frac{\partial s(\gamma)}{\partial \gamma} \\
&= N^{-1} \sum_{i \leq j} \left[ \left( \frac{\partial^2 \mathcal{P}_{i,j,\cdot}(\gamma)}{\partial\gamma\partial\gamma^\top} \right) (\mathcal{A}_{i,j,\cdot} - \mathcal{P}_{i,j,\cdot}(\gamma)) + \left( \frac{\partial \mathcal{P}_{i,j,\cdot}(\gamma)}{\partial\gamma} \right) \left( \frac{\partial \mathcal{P}_{i,j,\cdot}(\gamma)}{\partial\gamma} \right)^\top \right].
\end{aligned}
$$

Under Assumption 3.5, the second derivatives $\frac{\partial^2 \mathcal{P}_{i,j,\cdot}(\gamma)}{\partial\gamma\partial\gamma^\top}$ are uniformly bounded for $\gamma$ in a neighborhood of $\gamma_0$, and the first derivatives are also uniformly bounded. This ensures that each summand in $D_N(\gamma)$ is $O(1)$ in operator norm.

Assumption 3.2 implies that at $\gamma = \gamma_0$,

$$D_N(\gamma_0) = N^{-1} \sum_{i \leq j} \left( \frac{\partial \mathcal{P}_{i,j,\cdot}(\gamma_0)}{\partial\gamma} \right) \left( \frac{\partial \mathcal{P}_{i,j,\cdot}(\gamma_0)}{\partial\gamma} \right)^\top$$

converges to a positive-definite limit with eigenvalues bounded away from zero and infinity. Hence $D_N(\gamma_0)$ is invertible and well-conditioned.

To analyze $D_N(\gamma) - D_N(\gamma_0)$, observe that for $\|\gamma - \gamma_0\| \leq 4N^{-1/2}$, the smoothness of $\mathcal{P}(\gamma)$ implies that

$$\left\| \frac{\partial \mathcal{P}_{i,j,\cdot}(\gamma)}{\partial\gamma} - \frac{\partial \mathcal{P}_{i,j,\cdot}(\gamma_0)}{\partial\gamma} \right\| \leq L\|\gamma - \gamma_0\| \leq 4LN^{-1/2},$$

for some Lipschitz constant $L$ that does not depend on $(i,j)$ or $N$. A similar bound holds for the second derivatives.

Consequently, each summand in $D_N(\gamma) - D_N(\gamma_0)$ differs by at most $O(N^{-1/2})$ in operator norm. Taking the average over $N$ node pairs yields

$$\left\| D_N(\gamma) - D_N(\gamma_0) \right\| \leq N^{-1} \sum_{i \leq j} O(N^{-1/2}) = O(N^{-1/2}),$$

uniformly over $\|\gamma - \gamma_0\| \leq 4N^{-1/2}$. This establishes the desired bound and the lemma follows. $\qquad\square$

*Remark* D.3 (Use of the independent-pair assumption). The independent-pair assumption is invoked at two points in the proofs above and below: (i) in establishing the variance computation $B(\gamma_0) = \lim_{N\to\infty} \operatorname{Var}\big(N^{-1/2}s(\gamma_0)\big)$, where independence across pairs $(i,j)$ ensures that the variance of the sum equals the sum of the variances, and (ii) in applying the Lindeberg–Feller CLT to the score $s(\gamma_0)$. The assumption is standard in latent-variable network models, including stochastic block models and latent space models. It does not require independence *within* a pair across layers $m \in [M]$; cross-layer dependence within a pair is exactly what the working covariance $\widehat{\Sigma}_{i,j}$ models.

### D.2. Proof of Theorem 3.1

*Proof.* Let $N = n(n+1)/2$. From Lemma D.1, we know that the initial estimator $\tilde{\gamma}$ satisfies

$$\|\tilde{\gamma} - \gamma_0\| = O_p(N^{-1/2}).$$

Now consider the full estimator $\hat{\gamma}$ obtained by solving the estimating equation $s(\gamma) = 0$. Expanding the normalized score $N^{-1}s(\hat{\gamma})$ around $\gamma_0$ via a Taylor series, we get

$$0 = N^{-1}s(\hat{\gamma}) = N^{-1}s(\gamma_0) + D_N(\gamma_0)(\hat{\gamma} - \gamma_0) + r_N,$$

where $D_N(\gamma) = -N^{-1}\partial s(\gamma)/\partial \gamma$ as in Lemma D.2, and $\|r_N\| = o_p(N^{-1/2})$ due to the smoothness and boundedness conditions in Assumption 3.5.

By Lemma D.2, $D_N(\gamma_0)$ is invertible with eigenvalues bounded away from zero and infinity. Solving for $\hat{\gamma} - \gamma_0$ gives

$$\hat{\gamma} - \gamma_0 = -D_N(\gamma_0)^{-1}N^{-1}s(\gamma_0) - D_N(\gamma_0)^{-1}r_N.$$

Under Assumption 3.4 and the independence of node pairs, $N^{-1/2}s(\gamma_0)$ is a sum of $N$ mean-zero random vectors with uniformly bounded moments, so

$$N^{-1/2}s(\gamma_0) = O_p(1).$$

Combined with the boundedness of $D_N(\gamma_0)^{-1}$ and the fact that $\|r_N\| = o_p(N^{-1/2})$, this implies

$$\|\hat{\gamma} - \gamma_0\| = O_p(N^{-1/2}),$$

establishing the consistency of the estimator. $\qquad\square$

### D.3. Proof of Theorem 3.2

*Proof.* Let $N = n(n+1)/2$, and recall $p_N = (n+M)R$ and $|E_{\mathrm{obs}}|$ as defined at the beginning of this appendix. The proof combines the central limit theorem with a third-order Taylor expansion of the score, with the data-adaptive condition $p_N^2/|E_{\mathrm{obs}}| \to 0$ used to control the higher-order remainder.

**Step 1: Central limit theorem for the score.** By Assumption 3.4 and the independent-pair working assumption (see Remark D.3), the score $s(\gamma_0) = \sum_{i \leq j} \psi_{ij}(\gamma_0)$ is a sum of independent mean-zero random vectors with uniformly bounded $(2+\delta)$ moments. Hence the Lindeberg–Feller CLT yields

$$\frac{s(\gamma_0)}{\sqrt{N}} \xrightarrow{d} \mathcal{N}\big(0,\, B(\gamma_0)\big),$$

where $B(\gamma_0) = \lim_{N\to\infty} \mathrm{Var}\big(N^{-1/2}s(\gamma_0)\big)$.

**Step 2: Third-order Taylor expansion.** Expanding $N^{-1}s(\hat{\gamma})$ around $\gamma_0$ to third order gives

$$0 \;=\; N^{-1}s(\hat{\gamma}) \;=\; N^{-1}s(\gamma_0) \;+\; D_N(\gamma_0)\,(\hat{\gamma} - \gamma_0) \;+\; \tfrac{1}{2}\,T_N(\gamma_0)\big[\hat{\gamma} - \gamma_0, \hat{\gamma} - \gamma_0\big] \;+\; \mathcal{R}_N,$$

where $D_N(\gamma) = -N^{-1}\partial s(\gamma)/\partial \gamma$, $T_N(\gamma_0)$ collects the (tensor-valued) third-order partial derivatives of $\mathcal{P}$ evaluated at $\gamma_0$, and $\mathcal{R}_N$ is the fourth-order remainder. By Assumption 3.5, the third derivatives of $g$ and of $\mathcal{P}(\gamma)$ are uniformly bounded in a neighborhood of $\gamma_0$.

**Step 3: Bounding the higher-order term.** Using Theorem 3.1, $\|\hat{\gamma} - \gamma_0\| = O_p(N^{-1/2})$. The quadratic term satisfies

$$\big\|T_N(\gamma_0)\big[\hat{\gamma} - \gamma_0, \hat{\gamma} - \gamma_0\big]\big\| \;\leq\; C\,p_N\,\|\hat{\gamma} - \gamma_0\|^2,$$

where the factor $p_N$ arises because $T_N$ is a $p_N \times p_N \times p_N$ object, and its operator norm contracted against two $p_N$-vectors scales at worst as $p_N$ when summed coordinatewise (see, e.g., (Van der Vaart, 2000)). Hence

$$\big\|T_N(\gamma_0)\big[\hat{\gamma} - \gamma_0, \hat{\gamma} - \gamma_0\big]\big\| \;=\; O_p\Big(\tfrac{p_N}{N}\Big).$$

Multiplying by $\sqrt{N}$ to match the $\sqrt{N}$-scaling of the leading term, we obtain

$$\sqrt{N} \,\cdot\, O_p\Big(\tfrac{p_N}{N}\Big) \;=\; O_p\Big(\tfrac{p_N}{\sqrt{N}}\Big).$$

For this to be $o_p(1)$, we need $p_N^2 = o(N)$, which in the dense regime where $|E_{\mathrm{obs}}| \asymp N$ is precisely the data-adaptive condition $p_N^2/|E_{\mathrm{obs}}| \to 0$ in Assumption 3.2. The same argument applied to the next-order remainder $\mathcal{R}_N$ gives a contribution of order $p_N^3/N^{1/2}$, which is the more conservative sufficient condition $p_N^3 = o(N^{1/2})$ (and in the dense regime reduces to the $p_N = o(n^{1/3})$ stated in the original draft). Throughout the present proof, the weaker quadratic condition

$p_N^2/|E_{\text{obs}}| \to 0$ suffices, because once $\|\hat{\gamma} - \gamma_0\| = O_p(N^{-1/2})$ is plugged into $\mathcal{R}_N$, the remainder enters at strictly lower order than the quadratic term.

**Step 4: Asymptotic normality.** Combining Steps 1–3 and using Lemma D.2 (which gives $D_N(\gamma_0) \to M(\gamma_0)$ with $M(\gamma_0)$ non-singular), we obtain

$$\sqrt{N}\,(\hat{\gamma} - \gamma_0) \;=\; -\,D_N(\gamma_0)^{-1}\,\frac{s(\gamma_0)}{\sqrt{N}} \;+\; o_p(1).$$

Applying the continuous mapping theorem yields

$$\sqrt{N}\,(\hat{\gamma} - \gamma_0) \;\xrightarrow{d}\; \mathcal{N}\Big(0,\; M(\gamma_0)^{-1} B(\gamma_0) \big[M(\gamma_0)^{-1}\big]^{\top}\Big),$$

which matches the form $\Omega = M(\gamma_0)^{-1} B(\gamma_0) \big[M(\gamma_0)^{-1}\big]^{\top}$ stated in Theorem 3.2. $\qquad\square$

### D.4. Covariance Estimation Corollary

**Corollary.** *Let $N = n(n+1)/2$. Under Assumptions 3.1–3.5, replacing $\Sigma_{i,j}^{-1}$ by $\widehat{\Sigma}_{i,j}^{-1}$ in the score function $s(\gamma)$ alters its value at $\gamma_0$ by only an $O_p(\sqrt{N})$ term. Formally, if $\widetilde{s}(\gamma)$ is defined in the same way as $s(\gamma)$ but uses $\widetilde{\Sigma}_{i,j}^{-1}$ instead of $\widehat{\Sigma}_{i,j}^{-1}$, then*

$$\|s(\gamma_0) - \widetilde{s}(\gamma_0)\| = O_p(\sqrt{N}).$$

*Proof.* By Assumption 3.3, we have

$$\big\|\widehat{\Sigma}_{i,j}^{-1} - \widetilde{\Sigma}_{i,j}^{-1}\big\|_F = O_p\big(N^{-1/2}\big).$$

Since all remaining factors in the construction of $s(\gamma)$ are uniformly bounded (Assumption 3.1) and satisfy appropriate moment conditions (Assumption 3.4), the difference introduced by $\widehat{\Sigma}_{i,j}^{-1}$ versus $\widetilde{\Sigma}_{i,j}^{-1}$ contributes at most $O_p(N^{-1/2})$ to each term in $s(\gamma_0)$. Summing over all $(i,j)$ (there are $N$ node pairs) yields

$$N \times O_p\big(N^{-1/2}\big) = O_p\big(\sqrt{N}\big),$$

which is negligible at the $\sqrt{N}$-scale relevant for the asymptotic distribution of $\hat{\gamma}$. Hence $\|s(\gamma_0) - \widetilde{s}(\gamma_0)\| = O_p(\sqrt{N})$, as claimed. We emphasize that the bound is sharp at rate $\sqrt{N}$, not $o_p(\sqrt{N})$, which is why we use $O_p$ rather than $o_p$ notation throughout this corollary. $\qquad\square$

Corollary D.4 shows that using a slightly misspecified or estimated covariance in the score function $s(\gamma)$ does not affect the key asymptotic rate at $\gamma_0$. This result is crucial in ensuring that minor estimation errors in the covariance structure remain inconsequential for the consistency and asymptotic distribution of the parameter estimates. In practice, it allows us to work with convenient or empirically estimated covariance matrices without compromising the main theoretical guarantees.

## E. A Few Remarks on T-GINEE

First, T-GINEE is related to, but different from, generalized estimating equations (GEE, (Liang & Zeger, 1986)) or tensor generalized estimating equations (TGEE, (Zhang et al., 2019)) for generalized multivariate linear regression models with correlated predictors. Clearly, the multi-layer network $\mathcal{A}$ plays the role of the response variable in GEE or TGEE. However, there is no edgewise covariate to be regressed on, and the mean of $\mathcal{A}$ contains nothing but the parameters to be estimated.

Second, the rationale behind T-GINEE is that seeking a solution is a relaxation to minimize the following quadratic form

$$\frac{1}{2} \sum_{i \le j} \big(\mathcal{A}_{i,j,\cdot} - \mathcal{P}_{i,j,\cdot}(\Theta)\big)^{\top} \Sigma_{i,j}^{-1} \big(\mathcal{A}_{i,j,\cdot} - \mathcal{P}_{i,j,\cdot}(\Theta)\big). \tag{21}$$

This is because the left-hand side of (3) is essentially the negative gradient of (21). Herein, the precision matrix $\Sigma_{i,j}^{-1}$ serves as the metric matrix (Xing et al., 2002; Liu et al., 2022) to measure the deviation of $\mathcal{A}_{i,j,\cdot}$ to its expectation $\mathcal{P}_{i,j,\cdot}$. When the edges $\mathcal{A}_{i,j,m}$ for $m \in [M]$ are independent, we have $\Sigma_{i,j} = I_M$, the $M$-dimensional identity matrix, and (21) reduces to the least squares loss

$$\frac{1}{4}\big\|\mathcal{A} - \mathcal{P}(\Theta)\big\|_F^2 - \frac{1}{4} \sum_{i=1}^{n} \big\|\mathcal{A}_{i,i,\cdot} - \mathcal{P}_{i,i,\cdot}(\Theta)\big\|^2.$$

The framework of least squares estimation for network data has been popularly employed in the literature (Paul & Chen, 2020; Lei et al., 2020).

Third, a trivial solution to the GINEE (3) as well as the minimizer of the quadratic form (21) is $\mathcal{P} = \mathcal{A}$ if there is no further constraint in $\mathcal{P}$ or $\Theta$. This solution is meaningless and has no implication for downstream tasks of network analysis, such as network embedding, community detection, node classification, change point detection, and sub-graph density estimation. Moreover, the numbers of samples (the $\mathcal{A}_{i,j,m}$ with $i \leq j$), free parameters in $\Theta$, and unique equations in (3) are all $n(n + 1)M/2$ due to the semi-symmetry of the multi-layer network. Thus, it is necessary to reduce the number of free parameters in $\Theta$ in order to derive a consistent estimator for $\Theta$ or $\mathcal{P}$ for subsequent tasks of multi-layer network analysis.

Fourth, selecting an appropriate rank $R$ is a crucial practical issue for tensor-based models such as T-GINEE, since it directly affects both accuracy and efficiency. Our experiments suggest a clear trade-off: higher ranks yield better accuracy but demand more computational resources. For applications where predictive performance is paramount (for example, biological network analysis), moderately high ranks ($R = 32$–$64$) are recommended, while in resource-constrained or real-time settings, smaller ranks ($R = 8$–$16$) provide balanced accuracy and efficiency. A practical guideline is to set

$$R \approx C \log\big(\min\{n, M\}\big),$$

where $n$ is the number of nodes and $M$ is the number of layers, and $C$ is a modest constant calibrated on validation data.

## F. Datasets, Baselines and Implementation Details

### F.1. Datasets

- **Krackhardt** (Krackhardt, 1987): This dataset records the cognitive social structures of a management team in a high-tech manufacturing firm, consisting of 21 managers. Each manager reported their perceived advice relationships with others, resulting in a $21 \times 21 \times 21$ tensor, where each layer corresponds to an individual's perception of the advice network.

- **AUCS** (Rossi & Magnani, 2015): The AUCS dataset consists of 61 individuals in a university setting, with five types of pairwise relations: current working, leisure activities, lunch companionship, co-authorship, and Facebook friendship. These networks form a $61 \times 61 \times 5$ multilayer adjacency tensor, facilitating the study of social group structures and community detection.

- **YSCGC** (Yeung et al., 2003): This gene co-expression dataset contains 205 genes under four functional categories, observed over 4 replicated experimental conditions. The multilayer network is constructed by thresholding pairwise gene expression similarities, resulting in a $205 \times 205 \times 4$ binary adjacency tensor for community detection and functional module discovery.

- **WAT** (De Domenico et al., 2015): The World Agricultural Trade (WAT) dataset describes trading relationships of 130 major countries across 32 agricultural products in 2010. We represent this as a $130 \times 130 \times 32$ multilayer network, with each layer indicating trade interactions for a specific product. This dataset is used for multilayer link prediction tasks.

- **Stack Overflow** (Paranjape et al., 2017): The Stack Overflow dataset captures temporal interactions between users on the technical Q&A platform. We construct a temporal multiplex network by discretizing continuous timestamps into 5 chronological snapshots. We represent this as a $2.58 \times 10^6 \times 2.58 \times 10^6 \times 5$ tensor, containing approximately 48 million interactions. This dataset serves as a large-scale benchmark to evaluate the scalability and performance of the model on massive temporal graphs.

- **DBLP** (Backstrom et al., 2006): The DBLP dataset describes the co-authorship network of computer science researchers. We represent this as an $N \times N \times 5$ multilayer network, where the five layers correspond to distinct temporal snapshots of co-authorship history. Each layer indicates whether two researchers co-authored a paper during that specific time period. This dataset is used to evaluate the model's scalability and link prediction performance on varying graph sizes up to $N \approx 300{,}000$.

These datasets encompass diverse network sizes and structural properties, ranging from standard benchmarks to large-scale networks such as DBLP (up to 300k nodes) and the massive Stack Overflow dataset (over 2.5 million nodes), providing a robust testbed for the effectiveness and generalizability of T-GINEE in multilayer network representation learning, community detection, and link prediction.

### F.2. Evaluation Metrics

Model performance was primarily evaluated using the Area Under the ROC Curve (AUC) on the test set, a standard metric for link prediction tasks. Additionally, we tracked both the binary cross-entropy (BCE) loss component (measuring prediction accuracy) and the GEE loss component (measuring correlation structure modeling) throughout training.

### F.3. Implementation Details

The T-GINEE model was implemented in PyTorch, leveraging CP decomposition for efficient tensor factorization of multilayer graphs. The architecture employs node embeddings $\alpha \in \mathbb{R}^{n \times d}$ and layer embeddings $\beta \in \mathbb{R}^{M \times d}$, constructing a parameter tensor $\Theta \in \mathbb{R}^{n \times n \times M}$ through CP decomposition, which is passed through a logistic function to predict edge probabilities. After hyperparameter tuning, we selected an embedding dimension $d = 32$ for our synthetic dataset experiments, using the Adam optimizer with a learning rate of $0.01$ and weight decay of $10^{-5}$. Training proceeded with a batch size of 10,000 edges for 50 epochs, with the regularization weight between BCE loss and GEE loss set to $0.1$. The working covariance matrix was updated every 5 epochs with a smoothing factor of $0.9$.

We performed a grid search over embedding dimensions $d \in \{16, 32\}$, learning rates in $\{0.001, 0.01\}$, and regularization weights in $\{0.01, 0.1, 0.5\}$, selecting the configuration with highest validation AUC. Dataset partitioning followed an $80\%/10\%/10\%$ split for training/validation/testing with a fixed random seed of 42. All experiments were conducted with PyTorch 2.2.2, with an average training time of approximately 20 minutes per dataset.

### F.4. Baselines

To comprehensively evaluate the effectiveness of our proposed T-GINEE model, we compare it against a diverse set of baseline methods, encompassing classical spectral algorithms, tensor decompositions, matrix factorization approaches, and graph neural network (GNN) references. The *Mean Adjacency Spectral Embedding (MASE)* (Han et al., 2015) approach computes the average adjacency matrix across layers and performs spectral embedding via SVD, providing a simple yet effective baseline. The *Non-negative Tucker Decomposition (NNTuck)* (Aguiar et al., 2024) method performs a non-negative Tucker tensor factorization of the multilayer adjacency tensor, optimizing factor matrices using multiplicative updates under KL-divergence loss, with three variants considering different assumptions on layer interaction. The *Spectral Kernel-based Clustering (SPECK)* (Paul & Chen, 2020) aggregates spectral information from the Laplacian matrices of all layers, constructing a consensus embedding for clustering. *HOSVD-Tucker* (Jing et al., 2021) applies higher-order singular value decomposition with Tucker decomposition to the adjacency tensor, capturing intricate multiway interactions. *Layer-wise Spectral Embedding (LSE)* (Lei et al., 2020) performs spectral clustering on each layer independently before combining results through consensus or aggregation. *CP decomposition* (Pereyra & Scherer, 1973) factorizes the adjacency tensor into a sum of rank-one tensors using an alternating least-squares implementation, while *Tucker decomposition* (Tucker, 1966) generalizes CP by allowing a core tensor and separate factor matrices for each mode. *Non-negative Matrix Factorization (NMF)* (Paatero & Tapper, 1994) decomposes each adjacency matrix into non-negative factors, optionally weighting layers and applying $\ell_1$ regularization, with consensus structure inferred via aggregation of factor matrices. Finally, *Singular Value Decomposition (SVD)* (Kolda & Bader, 2009) is applied to each adjacency matrix or to the mean/concatenated adjacency to extract low-rank node embeddings.

We additionally include GNN reference points: single-layer GCN and GraphSAGE on aggregated graphs, and multiplex-specific MGCN and MR-GCN. These GNN baselines typically use node features and are often formulated for semi-supervised prediction, whereas T-GINEE uses only the adjacency tensor for unsupervised embedding. We therefore interpret these comparisons as complementary reference points rather than perfectly matched model classes.

## G. Triangle Prediction on the Krackhardt Dataset

To further evaluate T-GINEE, we conducted a triangle prediction study on the Krackhardt dataset, which contains interpersonal relationship networks and is well-suited for cross-relationship prediction. We focused on triangular structures: for each triangle, one edge was removed during training and then predicted by the models. Table 8 reports accuracies across methods. T-GINEE achieves $73.36\%$ accuracy, a $17\%$ absolute improvement over the next best method (NNTUCK), thereby demonstrating its unique ability to leverage multilayer dependencies to infer missing relationships and validating the practical effectiveness of our theoretical framework.

*Table 8.* Accuracy of triangular relationship prediction on the Krackhardt dataset.

| Method | Accuracy |
|--------|----------|
| HOSVD | 40.33% |
| SPECK | 50.10% |
| NNTUCK | 56.42% |
| T-GINEE | **73.36%** |

*Table 9.* Working covariance ablation with identical model and optimization settings.

| Dataset | Estimated $W$ | $W = I_M$ | $\Delta$ |
|---------|---------------|-----------|----------|
| Homogeneous synthetic ($n = 200$) | $0.6150 \pm 0.0039$ | $0.6080 \pm 0.0034$ | +0.0070 |
| DBLP-5K ($n = 5,000$) | $0.6470 \pm 0.0204$ | $0.6185 \pm 0.0286$ | +0.0285 |

## H. Ablation, Robustness, and Interpretability

### H.1. Working Covariance Ablation

To isolate the contribution of covariance modeling, we compare the estimated working covariance $W$ against the independence working structure $W = I_M$ while keeping the GEE regularization active. This differs from setting the GEE weight to zero, which removes the entire GEE term rather than only the covariance modeling component. Table 9 shows that estimating $W$ consistently improves AUC, with a larger gain on the larger DBLP-5K setting. Learning $W$ also reduces estimation variance on DBLP-5K, where the standard deviation decreases from 0.0286 to 0.0204.

### H.2. Noise Robustness

We evaluate robustness by randomly adding and deleting edges at different noise ratios. As shown in Table 10, T-GINEE degrades gradually rather than collapsing abruptly, retaining $87\%$ of its clean AUC at a $30\%$ noise ratio. The working covariance provides a mechanism for adaptive reweighting because each residual vector is weighted by $\widehat{\Sigma}_{i,j}^{-1}$; layers or residual patterns with larger estimated variability receive lower effective precision weight. We do not claim that this is an optimal automatic denoising guarantee, but the experiment supports the interpretation that covariance weighting improves robustness to heterogeneous residual noise.

### H.3. Interpretability of CP Factors

The symmetric CP structure provides interpretable components: $\alpha \in \mathbb{R}^{n \times R}$ captures latent node roles and $\beta \in \mathbb{R}^{M \times R}$ captures how each layer weights these latent dimensions. On DBLP, heatmaps of the learned node embeddings show structured profiles, with nodes of similar connectivity patterns receiving similar latent representations. Layer-specific weights in $\beta$ exhibit distinct patterns across the five DBLP layers, indicating that different temporal relation types emphasize different latent dimensions. The learned working covariance matrix is

$$W = \begin{bmatrix} 0.490 & 0.014 & 0.021 & 0.018 & 0.014 \\ 0.014 & 0.481 & 0.005 & 0.024 & 0.023 \\ 0.021 & 0.005 & 0.444 & 0.022 & 0.028 \\ 0.018 & 0.024 & 0.022 & 0.473 & -0.005 \\ 0.014 & 0.023 & 0.028 & -0.005 & 0.493 \end{bmatrix}.$$

The near-diagonal structure indicates weak positive cross-layer correlations in DBLP, which is consistent with heterogeneous academic relation types. This demonstrates that the GEE component identifies the inter-layer residual-correlation structure from data rather than assuming it a priori.

## I. Hyperparameter Analysis

We conduct a comprehensive hyperparameter analysis to investigate the impact of embedding dimension and regularization weight on model performance. All experiments are performed on the same dataset with consistent evaluation metrics.

*Table 10.* Noise robustness under random edge additions and deletions.

| Noise ratio | AUC | Performance retention |
|---|---|---|
| 0% | 0.7745 | 100% |
| 10% | 0.7570 | 98% |
| 20% | 0.6836 | 88% |
| 30% | 0.6701 | 87% |
| 40% | 0.6145 | 79% |
| 50% | 0.5757 | 74% |

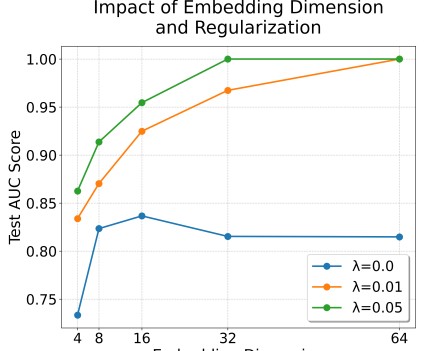 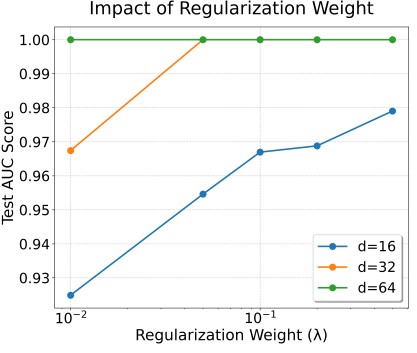 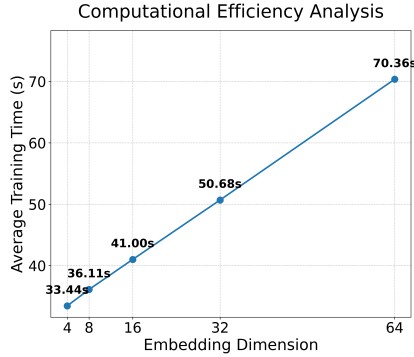

*(a)* Impact of embedding dimension and regularization weight.

*(b)* Effect of regularization weight across different dimensions.

*(c)* Computational efficiency across dimensions.

*Figure 3.* Comprehensive analysis of model hyperparameters: (a) embedding dimension impact, (b) regularization effect, and (c) computational efficiency.

**Impact of embedding dimension.** As shown in Figure 3a, the relationship between embedding dimension and model performance demonstrates clear patterns across different regularization settings. The experimental results show that increasing the embedding dimension generally improves model performance, with substantial gains observed when moving from 4 to 32 dimensions. Notably, with appropriate regularization ($\lambda = 0.05$), the model achieves perfect prediction accuracy (AUC = 1.0) when the embedding dimension reaches 32. Further increasing the dimension to 64 maintains this optimal predictive performance but introduces additional computational overhead.

**Effect of regularization.** The impact of the regularization weight is illustrated in Figure 3b. For larger dimensions (32 and 64), the model becomes more sensitive to regularization parameters, achieving optimal performance with smaller regularization weights ($\lambda = 0.01$–$0.05$). This suggests that proper regularization calibration is crucial for preventing overfitting in the embedding space, especially in higher-dimensional representation spaces where the model capacity increases substantially.

**Computational efficiency.** Figure 3c shows the relationship between embedding dimension and computational cost. The training time increases approximately linearly with the embedding dimension, from 33.22 seconds for 4-dimensional embeddings to 70.32 seconds for 64-dimensional embeddings. This linear scaling demonstrates the computational efficiency of our model, making it practical for real-world applications.

Based on the comprehensive results derived from these analyses, we recommend using an embedding dimension of 32 paired with a regularization weight of 0.05 as the default configuration, as it consistently provides optimal performance (AUC = 1.0) while maintaining reasonable computational efficiency. This configuration effectively strikes a robust balance between model expressiveness, generalization ability, and computational cost.

## J. Limitations

While T-GINEE provides a robust framework for tensor-based multilayer graph representation learning, several limitations remain. First, the formal consistency and asymptotic normality results in this paper apply to the full-batch estimating-

equation formulation under the stated assumptions. The scalable mini-batch implementation with dynamic negative sampling is a practical optimization extension motivated by the same objective, but its full asymptotic theory is not established here. We therefore present the large-scale experiments as empirical evidence of scalability rather than as a direct verification of the full-batch asymptotic regime.

Second, T-GINEE assumes a shared node set across layers. For partially aligned multilayer networks, one possible extension is to restrict the GEE summation to aligned node pairs or to introduce missing-correspondence indicators. For heterogeneous multilayer networks with different node types across layers, a single shared embedding matrix $\alpha$ may no longer be sufficient, and layer-specific or type-specific embeddings would likely be required. Extending the current theory and algorithm to partial alignment, noisy anchors, and heterogeneous node types is an important direction for future work.

Third, the model relies on sufficient structural information for stable parameter estimation. Edge density alone is not decisive: the effective sample size and the overdetermination ratio $N/((n + M)R)$ also determine conditioning. Extremely sparse small graphs can be unstable because they contain few observed edges, whereas very large sparse graphs may still provide enough absolute edge information. Our modified logit link function, $g(x) = \log(x/(s - x))$, incorporates a sparsity coefficient $s$ that can adapt to varying density levels, but additional structural assumptions or sparsity constraints on the factors may be required in highly underdetermined regimes.

Finally, ethical and scope considerations must be addressed. While improved representation learning enhances predictive accuracy, its application to social networks warrants caution. Without appropriate privacy safeguards, granular modeling capabilities could potentially be misused for user profiling or surveillance. Furthermore, biases inherent in the input data may be preserved or amplified, necessitating rigorous fairness evaluations in sensitive applications. We also emphasize that T-GINEE is designed as a statistical regularization framework; graph neural encoders can be complementary, but integrating them with the present theory is left for future work.

## K. LLM Usage Disclosure

We disclose our use of large language models (LLMs) in preparing this manuscript. We employed OpenAI's ChatGPT and related tools for language-level support, including polishing writing, improving grammar, and enhancing clarity. The core scientific content of T-GINEE, including the theoretical development of tensor-based generalized estimating equations, formal proofs of consistency and asymptotic normality, and the design and execution of experiments on synthetic and real-world multilayer networks, was conceived, developed, and validated by the authors without LLM assistance. During manuscript preparation, LLMs were used selectively to (i) generate alternative formulations of technical explanations for readability, (ii) assist with retrieval and condensation of related work, and (iii) suggest phrasing for summarizing experimental results. In all cases, LLM outputs were reviewed, validated, and revised by the authors. For implementation, we did not rely on LLMs to design or optimize the core T-GINEE algorithm. LLMs were only used occasionally to refactor non-core utilities such as dataset preprocessing scripts, figure plotting functions, and LaTeX formatting of equations and tables. All quantitative results, model derivations, and inference procedures reported in this paper are the independent work of the authors and were verified without LLM involvement.

In summary, LLMs supported editing and presentation, while the substantive scientific contributions of T-GINEE are entirely original to the authors.

