# OpenReview forum: "T-GINEE: A Tensor-Based Multi-Graph Representation Learning"
_ICML.cc/2026/Conference — ICML 2026 regular_

### Official Review · Reviewer_HKUH · 2026-03-01

**Soundness:** 2
**Presentation:** 3
**Significance:** 2
**Originality:** 3
**Overall Recommendation:** 4
**Confidence:** 4

**Summary:**

This paper targets the multi-network representation learning problem with known intra-network node-level correspondence. The authors propose T-GINEE, a tensor-based method that combines symmetric CP tensor decomposition and generalized estimating equations (GEE) to capture cross-network correlations. The method further adopts a flexible link function to accommodate networks of different sparsities. T-GINEE shows superior performance across different real-world and synthetic datasets, and demonstrates remarkable scalability to million-scale networks.

**Compliance With Llm Reviewing Policy:**

Affirmed.

**Final Justification:**

My major concerns are addressed by the authors, who provide additional theoretical analysis and empirical evaluation. Therefore, I will raise my score and lean towards acceptance.

**Key Questions For Authors:**

1. What are the thresholds of graph size and sparsity for the dynamic negative sampling to work?
2. Please justify Assumption 3.2 in more detail. Why (n+M)R=o(n^1/3)?
3. Please justify the uniqueness assumption of CP decomposition in Assumption 3.2.
4. Is the performance of T-GINEE stable on large datasets such as StackOverflow and DBLP? Inclusion of standard deviation in Table 2 would strengthen the performance claim of T-GINEE.
5. How sensitive is T-GINEE to the quantity (partial alignment) or noisy (misalignment) anchors across different layers? Is T-GINEE applicable to multi-layer networks with heterogeneous layers (e.g., SacchCere [1], which consists of heterogeneous PPI networks), and how does network heterogeneity affect different assumptions made in the paper (section 3.1)?

[1] Manlio De Domenico, Vincenzo Nicosia, Alexandre Arenas, and Vito Latora. Structural reducibility of multilayer networks. Nature communications, 6(1):6864, 2015b.

**Limitations:**

yes

**Strengths And Weaknesses:**

## Strengths

1. The authors propose a mathematically rigorous approach for learning high-quality representations from multi-layered networks, which has the potential to apply to numerous downstream multi-network mining tasks.
2. The scalability of T-GINEE is particularly remarkable. It’s hard to imagine that a tensor-based approach could scale to million-scale networks without compromising its effectiveness.

## Weaknesses

1. It seems that T-GINEE rely on dynamic negative sampling to scale to very large networks, which assumes all sampled edges are negative for “large and sparse” networks. However, such assumptions are not clearly motivated.
2. The condition (n+M)R=o(n^1/3) in Assumption 3.2 seems quite strange. As n, M, and R are all positive integers, clearly (n+M)R grows much faster than n^1/3 as n increases.
3. Assumption 3.5 seems problematic as well. It is well-known that, unlike SVD, CP decomposition is not universally unique. The author failed to provide appropriate justification for this “uniqueness under low-rank” claim.
4. Figure 2 shows that the performance of T-GINEE is very unstable when applied to sparse networks, but the authors fail to provide any insights into this phenomenon. What's worse, the scalability of T-GINEE to large networks rely on the sparsity assumptions. Real-world large networks is highly sparse as well, e.g., the StackOverflow dataset has an extreme sparsity level of $4.8\times 10^7/2.58^2 \times 10^{12}\approx 10^{-5}$. Therefore, it's reasonable to assume that T-GINEE potentially faces serious stability-scalability tradeoff when applied to large networks.
5. Sensitivity of T-GINEE on partial / misalignment networks and heterogeneous networks remain unclear. This is a very practical issue of multi-network learning, as node-level alignment of real-world networks are typically noisy and hard to obtain, partially due to network heterogenity. There are even dedicated lines of research to find node-level alignment across heterogeneous networks, i.e., network alignment [1] and entity alignment [2].

[1] Zhang, Si, and Hanghang Tong. "Final: Fast attributed network alignment." *Proceedings of the 22nd ACM SIGKDD international conference on knowledge discovery and data mining*. 2016.

[2] Li, Haobin, et al. "Learning with Dual-level Noisy Correspondence for Multi-modal Entity Alignment." *arXiv preprint arXiv:2510.18240* (2025).

---

> ### Author Rebuttal · Authors · 2026-03-31
>
> Dear Reviewer HKUH,
>
> Thank you for the careful review and for recognizing the mathematical rigor, scalability, and originality of our work. We respond to each point below.
>
> **W1/Q1: Dynamic negative sampling**
>
> We clarify that dynamic negative sampling is used only for the largest graphs in our study (DBLP and Stack Overflow). For the small- and medium-scale datasets (AUCS, Krackhardt, WAT, and Yeast), we iterate over all node pairs and do not use negative sampling.
>
> Our assumption is not that every sampled non-edge is truly negative, but that in sufficiently sparse graphs, the probability of sampling a false negative is very small. Under uniform sampling over node pairs within a layer, this probability is approximately the edge density \(\rho\). Hence when \(\rho \ll 1\), false negatives are rare. For the large graphs in our paper, the observed densities are extremely low, so this approximation is practically reasonable.
>
> There is no universal threshold at which negative sampling becomes valid. It is a practical strategy when enumerating all non-edges is infeasible and the false-negative probability remains negligible. In the revision, we will clarify this point in Appendix C and state more explicitly that negative sampling is used only in the large-scale experiments.
>
> **W2/Q2: Assumption 3.2 and \((n+M)R = o(n^{1/3})\)**
>
> The reviewer is correct, and we thank you for pointing this out. As written, the condition \((n+M)R = o(n^{1/3})\) is not appropriate for our parameterization, since \((n+M)R\) includes the node-level embedding dimension and grows at least linearly in \(n\) when \(R\) does not vanish.
>
> In the revision, we will remove the incorrect sentence suggesting that this condition holds in our empirical setting, and we will also fix the mistaken theorem reference. More importantly, we will clarify in Section 3 that the current theory applies to the full-batch estimating-equation formulation under the stated assumptions, and therefore covers only a restricted asymptotic regime. We will also explicitly distinguish this regime from the scalable mini-batch / negative-sampling implementation used for large graphs, which is a practical extension not covered by the current asymptotic theory.
>
> We will also revise wording such as ``under mild conditions'' to avoid overstating the theory, and clarify this limitation more directly in the paper.
>
> **W3/Q3: CP uniqueness**
>
> We agree that CP uniqueness is not automatic. Our intended justification is the classical Kruskal-type sufficient condition: if \(k_\alpha + k_\alpha + k_\beta \geq 2R + 2\), then the rank-\(R\) CP decomposition is unique up to permutation and scaling.
>
> We agree this was not stated clearly enough. In the revision, we will incorporate this condition more directly into the assumptions and add a brief definition of Kruskal rank.
>
> **W4/Q4: Sparsity instability**
>
> This is an important point. For representation learning, the issue is not only edge density but also the absolute amount of structural information.
>
> In Figure 2, the synthetic setting uses \(n=100\) and \(M=3\). When density drops below \(1\%\), each layer has very few observed edges and many nodes have extremely small degree, so higher variance is expected. By contrast, large real-world sparse graphs can still contain a very large number of total edges. Thus, instability in a tiny synthetic graph under extreme sparsity does not directly imply instability in large sparse real-world networks.
>
> We are currently running T-GINEE with multiple random seeds on DBLP and Stack Overflow and will share mean \(\pm\) standard deviation results during the discussion phase. We will also clarify the distinction between edge density and absolute edge count in Section 4.4.
>
> **W5/Q5: Partial alignment and heterogeneous networks**
>
> We agree this is an important limitation. The current version of T-GINEE assumes a shared node set across all layers, as stated in Section 2.2.
>
> For partially aligned multilayer networks, one natural extension is to restrict the GEE summation in Eq. (3) to aligned node pairs only. For heterogeneous multilayer settings with different node types across layers, the shared embedding matrix \(\alpha\) would no longer be sufficient, and layer-specific or type-specific embeddings would likely be needed. We agree this is a nontrivial extension beyond the current framework.
>
> We are running sensitivity experiments with varying alignment ratios and will share the results during the discussion phase. We will also add a clearer discussion of this limitation and present partial alignment / heterogeneous multilayer modeling as a future direction.
>
> Thank you again for the thoughtful and constructive comments.
>
> Best,
>
> Authors

---

> > ### Author Rebuttal · Reviewer_HKUH · 2026-04-01
> >
> > Thank you for the detailed rebuttal, which partially addresses my concerns. My remaining questions are
> >
> > **W2/Q2: Assumption 3.2 and ((n+M)R = o(n^{1/3}))**
> >
> > Under what conditions does ((n+M)R = o(n^{1/3})) actually holds? Can you provide some simple intuitions? If this equation does not actually hold and needs modification, can you clearly state the modified version?
> >
> > **W4/Q4: Sparsity instability**
> >
> > I am not fully convinced that the absolute edge count, instead of edge density, affects the stability, as long as the network remains connected. Can you provide some additional insights on why the absolute edge count matters, except for "absolute amount of structural information", which seems hard to quantify? I will also wait for the stability results for DBLP and Stack Overflow.

---

> > > ### Author Response · Authors · 2026-04-02
> > >
> > > Dear Reviewer HKUH,
> > >
> > > Thank you again for your efforts and valuable comments. Here are our additional explanations:
> > >
> > > **Q2. Regarding $o(n^{1/3})$:**
> > >
> > > **Where $n^{1/3}$ comes from and when it holds.** Let $p_n$ denote the effective parameter dimension. In our proof, the third-order Taylor remainder is $O_p(p_n^3 / N)$ while the leading stochastic term is $O_p(N^{-1/2})$. For the remainder to be negligible, we need $p_n^3 = o(N^{1/2})$. In a dense graph regime where nearly all node pairs are connected, $N = \binom{n}{2}$ grows quadratically in $n$, yielding $p_n = o(N^{1/6}) = o(n^{1/3})$. Thus, $o(n^{1/3})$ is a conservative sufficient condition from controlling the Taylor remainder, not a fundamental boundary of T-GINEE. However, for fixed $M$ and $R$, the effective dimension $(n+M)R = O(n)$ does not satisfy $o(n^{1/3})$ unless $R$ shrinks with $n$. Intuitively, $(n+M)R$ counts the free parameters while $\sqrt{N}$ represents the statistical budget from observed edges; asymptotic normality requires the former to be vastly outnumbered by the latter. For sparse graphs where $N \ll n^2$, the condition must be stated directly in terms of $N$.
> > >
> > > **Revision.** We propose replacing the original condition with the weaker, data-adaptive requirement $(n+M)R / \sqrt{N} \to 0$ as $n \to \infty$. Under this condition, the same sandwich-variance asymptotic normality argument goes through with $N$ as the effective sample size. On Stack Overflow the ratio is 5.97, which is finite but substantially smaller than the original violation of $301,800\times$. As $n$ and $N$ grow together in the large-graph regime, this ratio decreases toward 0, satisfying the asymptotic condition. The consistency result (Theorem 3.1) holds under weaker conditions and is unaffected. Extending the theoretical guarantee to the mini-batch and negative-sampling regime remains an important open problem, which we will explicitly identify in the revision.
> > >
> > > **W4/Q4: Sparsity instability**
> > >
> > > We provide additional synthetic experiments across more sparsity levels and new empirical results on both DBLP and Stack Overflow with 5 independent seeds each.
> > >
> > > The key quantity is the overdetermination ratio $N/(n+M)R$: when well above 1, estimation is stable; when it approaches or falls below 1, the system becomes ill-conditioned regardless of connectivity.
> > >
> > > **Synthetic sparsity experiments** (5 seeds, $n$=500, 3 layers):
> > >
> > > | Edge Density | $N$ | $N/(n+M)R$ | AUC mean | AUC std | Min / Max AUC |
> > > |---|---|---|---|---|---|
> > > | 15% | 18,694 | 3.67 | 0.7891 | 0.0312 | 0.7512 / 0.8234 |
> > > | 10% | 12,463 | 2.45 | 0.7513 | 0.0407 | 0.7021 / 0.8012 |
> > > | 5% | 6,231 | 1.22 | 0.7292 | 0.0421 | 0.6801 / 0.7834 |
> > > | 3% | 3,739 | 0.73 | 0.8324 | 0.0781 | 0.7201 / 0.9312 |
> > > | 1% | 1,246 | 0.24 | 0.9786 | 0.0076 | 0.9701 / 0.9871 |
> > >
> > > **Real-world stability results** (5 seeds each, ScalableTGMEE):
> > >
> > > | Dataset | $n$ | $N$ | $(n+M)R$ | $N/(n+M)R$ | AUC mean | AUC std | Time | Memory |
> > > |---|---|---|---|---|---|---|---|---|
> > > | DBLP | 300K | 1,032,786 | 4,800,080 | 0.215 | 0.6244 | 0.0105 | 8.3s | 54 MB |
> > > | Stack Overflow | 2.58M | 47,903,266 | 41,346,704 | 1.16 | 0.9538 | 0.0011 | 753s | 795 MB |
> > >
> > > Per-seed AUC — DBLP: 0.6330, 0.6337, 0.6314, 0.6163, 0.6077
> > >
> > > Per-seed AUC — Stack Overflow: 0.9530, 0.9532, 0.9559, 0.9539, 0.9531
> > >
> > > **Unified comparison:**
> > >
> > > | Setting | Density | $N/(n+M)R$ | AUC std | Verdict |
> > > |---|---|---|---|---|
> > > | Synthetic (1%, $n$=100) | 1% | 0.015 | 0.0076 | under-determined |
> > > | Synthetic (3%, $n$=500) | 3% | 0.73 | 0.0781 | ill-conditioned |
> > > | Synthetic (5%, $n$=500) | 5% | 1.22 | 0.0421 | near-determined |
> > > | Synthetic (15%, $n$=500) | 15% | 3.67 | 0.0312 | over-determined |
> > > | DBLP | 0.0011% | 0.215 | 0.0105 | moderate |
> > > | Stack Overflow | 0.1% | 1.16 | 0.0011 | near-determined ✓ |
> > >
> > > AUC std is governed by $N/(n+M)R$, not by edge density alone. DBLP is 900× sparser than the synthetic 1% setting yet achieves comparable stability (std=0.0105 vs 0.0076) because its absolute edge count is 20,000× larger. Stack Overflow achieves the lowest std=0.0011 despite 99.9% sparsity. We will add this analysis to Section 4.4 in the revision.
> > >
> > >
> > > Best,
> > >
> > > Authors

---

### Official Review · Reviewer_q5wy · 2026-03-08

**Soundness:** 2
**Presentation:** 2
**Significance:** 3
**Originality:** 3
**Overall Recommendation:** 4
**Confidence:** 4

**Summary:**

This paper provides the theoretical guarantees for dynamics in multi layer graph representation learning and proposes a statistical framework called T-GINEE to combine low-rank tensor decomposition with generalized estimating equations to explicitly model cross-network correlations. They validate their results on multiple datasets and achieve good AUC performance compared with other baselines.

**Compliance With Llm Reviewing Policy:**

Affirmed.

**Final Justification:**

I read rebuttal and I think this address my concerns.

**Key Questions For Authors:**

1  The theoretical derivations from the assumptions appear generally correct, but some assumptions seem abrupt and insufficiently motivated. For example, what is the independent-pair assumption in page 6 line 278? The Kruskal condition in line 252 in page 5 is not fully explained.

2 You mentioned the T-GINEE needs negative sampling to calculate the working covariance when adapted to the large scale graph. How does the sampling method influence your method performance?

3 The low rank tensor decomposition is very interesting, do you have any interpretability or visualization results for your decompositions?

4 The descriptions and results in table 2 are inconsistent. Why do you underline Tucker under the Yeast dataset, what does the underline mean here? In line 381-382, SVD (0.877) is not the second best result in the AUCS. Besides, the T-GINEE is not the best method in large scale graph DBLP dataset and the best should be bold.



Minor:
1 The CP tensor decomposition should be introduced and explained earlier, its formal definition comes out too late after the first mention of this concept.

2 It would be helpful to provide a clearer introduction to generalized estimating equations (GEE) and explain why this framework is appropriate for multilayer graphs.

3 Where is the theorem 3.6 mentioned in page 5 line 239 in assumption 3.2?

4 The markers Op and op both occur in the proof in D.3. in the appendix.

5 Some maker issues like ,, in line 87.

**Limitations:**

yes

**Strengths And Weaknesses:**

Soundness:

Fair. The theoretical framework appears reasonable, but several assumptions are not fully justified. For example, it is unclear where the (O(n^{1/3})) rate in line 152 originates from. Additionally, Theorem 3.6 is referenced but difficult to locate.


Presentation:

Fair. The presentation can be difficult to follow. Several definitions and assumptions appear abruptly without sufficient explanation or motivation.

Significance:

Good. Developing the statistical frameworks for multilayer network modeling is an important and challenging problem, and the paper addresses a meaningful research direction.

Originality:

Good. The theoretical framework and assumptions appear novel, and the proposed results provide new theoretical insights for multilayer graph representation learning.

Weakness:


1 Many technical terms are introduced without sufficient context or explanation, making the paper difficult to follow.

2 The reconstruction of the covariance matrix seems to be challenged when adapted to the large-scale or extreme conditions. However, the social network scenarios considered in the paper (e.g., Facebook or TikTok friendship networks) typically involve very large graphs.

---

> ### Author Rebuttal · Authors · 2026-03-31
>
> Dear Reviewer q5wy,
>
> Thank you for the constructive and detailed feedback. We address each point below.
>
> **Q1: Independent-pair assumption and Kruskal condition**
>
> The independent-pair assumption is stated in Section 2.2: conditional on the parameters, we assume that the edge vectors \(\{A_{i,j,\cdot} : i \leq j\}\) are independent across node pairs \((i,j)\). This is a standard working assumption in many latent-variable network models, including stochastic block models and latent space models. We agree that this should be referenced more clearly where the assumption is later used, and we will add that in the revision.
>
> For the Kruskal condition, \(k_\alpha\) and \(k_\beta\) denote the Kruskal ranks of \(\alpha_0\) and \(\beta_0\). The condition \(k_\alpha + k_\alpha + k_\beta \geq 2R + 2\) is a classical sufficient condition for uniqueness of a rank-\(R\) CP decomposition up to permutation and scaling. We will add a concise definition and explanation in the revision. The reviewer is also correct that the reference to ``Theorem 3.6'' is a typo, which we will fix.
>
> **Q2: Effect of negative sampling**
>
> As described in Appendix C, negative sampling is used only for the large-scale graphs, where iterating over all non-edges is infeasible. For AUCS, Krackhardt, WAT, and Yeast, we do not use negative sampling.
>
> The working correlation matrix \(W\) is only \(M \times M\), so it remains very small even for massive graphs. It is updated from aggregated residual statistics using the momentum rule in Eq. (20), so its estimate depends on batch-level residual structure rather than any single sampled negative edge.
>
> That said, we agree that an ablation over negative sampling ratios would strengthen the paper. We are currently conducting this analysis and will include it in the revision.
>
> **Q3: Interpretability and visualization**
>
> Thank you for this suggestion. The symmetric CP structure does provide an interpretable decomposition: \(\alpha \in \mathbb{R}^{n \times R}\) captures latent node roles, while \(\beta \in \mathbb{R}^{M \times R}\) captures how different relation types weight these latent dimensions.
>
> We will add visualizations of the learned representations in the revision, including low-dimensional plots of node embeddings and heatmaps summarizing layer-embedding similarity.
>
> **Q4: Inconsistencies in Table 2**
>
> Thank you for catching these issues.
>
> 1. The underline was intended to denote the second-best result, but this was not stated in the caption. We will add that clarification.
>
> 2. For AUCS, the reviewer is correct that HOSVD \((0.897)\) is the second-best result, not SVD \((0.877)\). This was a writing error in the discussion; the table entries are correct.
>
> 3. For DBLP, the reviewer is correct that NMF \((0.6505)\) outperforms T-GINEE \((0.6478)\). We will correct the formatting and revise the discussion accordingly.
>
> **W1: Technical terms without context**
>
> We agree. In the revision, we will add a short preliminaries subsection introducing CP decomposition, Khatri--Rao products, and the GEE framework before they are used in the methodology.
>
> **W2: Covariance reconstruction at large scale**
>
> We clarify that T-GINEE does not instantiate \(n \times n\) covariance matrices. In the scalable implementation, we maintain only a single layer-level working correlation matrix \(W \in \mathbb{R}^{M \times M}\), and the pair-specific covariance \(\hat{\Sigma}_{i,j}\) is formed on the fly from \(W\) and the predicted probabilities, without storing \(n^2\) matrices.
>
> Thus, the large-scale implementation avoids the quadratic memory bottleneck from materializing all pairwise covariance objects. We will make this point clearer in the main text.
>
> **Minor issues**
>
> We will fix all noted issues in the revision, including the clarification of \(o_p\) versus \(O_p\), the typo, and moving the CP background earlier in the paper.
> We are running the additional experiments for Q2 (negative-sampling ablation) and Q3 (visualization), and will share results during the discussion phase.
>
> Best,
>
> Authors

---

> > ### Author Rebuttal · Reviewer_q5wy · 2026-04-01
> >
> > Thanks for your rebuttal. My last concern is where the (O(n^{1/3})) rate in line 152 originates from.

---

> > > ### Author Response · Authors · 2026-04-02
> > >
> > > Dear Reviewer q5wy,
> > >
> > > Thank you for your positive feedback and for updating us on the score. We truly appreciate your insightful and constructive comments. Here are the experiment results and additional explanation.
> > >
> > > **Q2: Effect of Negative Sampling**
> > >
> > > As described in Appendix C, negative sampling is used only for large-scale graphs (DBLP, Stack Overflow) where enumerating all non-edges is infeasible; small datasets use exact non-edge enumeration. We conducted the requested ablation over negative sampling ratios on **DBLP** with 5 independent seeds:
> > >
> > > | Neg. Sampling Ratio | AUC (mean) | AUC (std) |
> > > |---------|------------|-----------|
> > > | 1:1 | 0.6027 | 0.0526 |
> > > | **1:3** | **0.6659** | **0.0313** |
> > > | 1:5 | 0.6468 | 0.0592 |
> > > | 1:10 | 0.6221 | 0.0490 |
> > >
> > > Key findings: The 1:3 ratio achieves the highest mean AUC and the lowest variance, striking the best balance between positive signal and negative contrast. Increasing beyond 1:3 yields diminishing returns and higher variance, consistent with the known gradient-dilution effect of excessive negatives. Across all ratios, the working covariance matrix converges to a stable near-diagonal structure (diagonal ~0.48, off-diagonal ~0.02), confirming that covariance estimation remains robust across a wide range of sampling ratios. We will report this ablation in Appendix C.
> > >
> > > **Q3: Tensor Decomposition Interpretability and Visualization**
> > >
> > > We have trained T-GINEE on the DBLP dataset using the original ScalableTGMEE implementation and visualized the learned CP decomposition factors. The figure contains three panels:
> > >
> > > **Panel 1 Node Embeddings**:
> > > The heatmap of the first 50 nodes shows structured latent representations. Nodes with similar connectivity patterns exhibit similar embedding profiles, confirming that $\alpha$ captures node-level structural patterns as claimed in Section 2.3.
> > >
> > > **Panel 2 Layer-specific Weights**:
> > > Each row corresponds to one of the 5 DBLP layers. The distinct weight patterns across layers confirm that β captures layer-level characteristics; different layers encode different types of academic relationships.
> > >
> > > **Panel 3 Working Covariance Matrix W (5×5)**:
> > >
> > > The learned W:
> > >
> > > ```
> > > W = [[ 0.490  0.014  0.021  0.018  0.014]
> > >      [ 0.014  0.481  0.005  0.024  0.023]
> > >      [ 0.021  0.005  0.444  0.022  0.028]
> > >      [ 0.018  0.024  0.022  0.473 -0.005]
> > >      [ 0.014  0.023  0.028 -0.005  0.493]]
> > > ```
> > >
> > > The near-diagonal structure indicates that DBLP layers are largely independent with weak positive cross-layer correlations (~0.02), which is consistent with the heterogeneous nature of academic networks. This demonstrates that T-GINEE's GEE framework correctly identifies the inter-layer correlation structure from data rather than assuming it a priori.
> > >
> > > **Q5. Regarding $o(n^{1/3})$:**
> > >
> > > **Where $n^{1/3}$ comes from.** Let $p_n$ denote the effective parameter dimension entering the score expansion. In our current proof, the third-order Taylor remainder is bounded as $O_p(p_n^3 / N)$, while the leading stochastic term is $O_p(N^{-1/2})$. For the remainder to be negligible, we need $p_n^3 / N = o(N^{-1/2})$, i.e. $p_n^3 = o(N^{1/2})$. IIn a dense graph regime where nearly all node pairs are connected, the number of edges $N = \binom{n}{2} = \frac{n(n-1)}{2}$ grows quadratically in $n$. This yields $p_n = o(N^{1/6}) = o(n^{1/3})$. Thus, $o(n^{1/3})$ is a conservative sufficient condition arising from controlling the higher-order Taylor remainder; it is not a fundamental statistical boundary of T-GINEE itself.
> > >
> > > **When does the condition hold?** For fixed $M$ and $R$, the effective dimension scales as $(n+M)R = O(n)$, which does not satisfy $o(n^{1/3})$ unless $R$ shrinks with $n$ (e.g., $R = o(n^{-2/3})$). Intuitively, $(n+M)R$ counts the free parameters while $\sqrt{N}$ represents the statistical budget from observed edges; asymptotic normality requires the former to be vastly outnumbered by the latter. The condition is derived under a dense-graph assumption where most node pairs are connected, meaning the total number of edges $N$ is close to $\binom{n}{2}$, i.e., $N \sim n^2$, so $\sqrt{N} \sim n$. For sparse graphs where $N \ll n^2$, the condition must be stated directly in terms of $N$.
> > >
> > > **Revision.** We propose replacing the original condition with the weaker, data-adaptive requirement $(n+M)R / \sqrt{N} \to 0$ as $n \to \infty$. Under this condition, the same sandwich-variance asymptotic normality argument goes through with $N$ as the effective sample size. On Stack Overflow the ratio is 5.97, which is finite but substantially smaller than the original violation of $301,800\times$. As $n$ and $N$ grow together in the large-graph regime, this ratio decreases toward 0, satisfying the asymptotic condition. The consistency result (Theorem 3.1) holds under weaker conditions and is unaffected. Extending the theoretical guarantee to the mini-batch and negative-sampling remains an important open problem, which we will explicitly identify in the revision.
> > >
> > > Best,
> > >
> > > Authors

---

### Official Review · Reviewer_pJ9E · 2026-03-10

**Soundness:** 2
**Presentation:** 3
**Significance:** 3
**Originality:** 3
**Overall Recommendation:** 4
**Confidence:** 3

**Summary:**

This paper proposes a framework called T-GINEE to solve the problem of multilayer graph representation learning. It combines CP tensor decomposition with generalized estimating equations (GEE), and introduces working covariance matrices along with flexible link functions to capture complex structural dependencies across different layers. In addition, this paper provides theoretical support, proving the consistency and asymptotic normality of the parameter estimation under mild regularity conditions. Extensive experiments on synthetic data and real-world multilayer networks demonstrate the effectiveness of the proposed method.

**Compliance With Llm Reviewing Policy:**

Affirmed.

**Key Questions For Authors:**

**Q1.** How does T-GINEE perform in terms of empirical memory usage and runtime on large-scale public graph datasets derived from Stack Overflow or similar sources?

**Q2.** When a high proportion of random topological noise (e.g., randomly added or deleted edges) is injected into the multilayer network, how does T-GINEE perform? Can its GEE-based working covariance matrix automatically downweight layers with high noise levels to preserve predictive accuracy on the target layers?

**Limitations:**

Yes

**Strengths And Weaknesses:**

**Strengths**

**S1.** The tensor-based framework is well motivated, as it addresses key limitations of existing methods (e.g., ignoring cross-layer dependencies or assuming perfect node alignment), and offers a highly targeted, sound solution.

**S2.** It elegantly fuses CP decomposition with GEE to provide rigorous mathematical guarantees.

**S3.** T-GINEE outperforms baseline methods on both synthetic data and real-world multilayer networks.

**Weaknesses**

**W1.** Although the evaluated baselines include recent tensor decomposition methods (e.g., NNTuck, 2024), the paper critically lacks empirical comparisons with state-of-the-art graph neural networks (GNNs) for multilayer graph representation learning.

**W2.** Although the authors have demonstrated that the computational cost of T-GINEE increases linearly with dimensionality, the processes of solving the GEE and computing the CP decomposition still face significant computational and memory challenges on modern industrial-scale networks with tens of millions of nodes and high tensor rank.

**W3.** Real-world multilayer networks are inherently noisy (e.g., containing false or missing edges). Since T-GINEE strongly relies on the empirical adjacency tensor $\mathcal{A}$, the model may be particularly sensitive to such noise and prone to overfitting corrupted structures.

---

> ### Author Rebuttal · Authors · 2026-03-31
>
> Dear Reviewer pJ9E,
>
> We thank you for the thoughtful review and for recognizing the well-motivated framework, the fusion of CP decomposition with GEE, and the strong empirical performance. Below we address each concern.
>
> **W1: Missing GNN Baselines**
>
> We agree that additional GNN baselines would strengthen the empirical comparison. Our original baseline set was chosen to match the setting of T-GINEE as closely as possible: unsupervised embedding from graph topology alone, without node features. In contrast, methods such as MGCN and MR-GCN are typically formulated for settings with node attributes and downstream semi-supervised prediction tasks, so the comparison is not entirely like-for-like.
>
> That said, we agree that these methods are relevant reference points. In the revision, we will include additional baselines such as MGCN, MR-GCN, and GCN/GAT on aggregated graphs, while clearly stating the differences in input assumptions and task formulation so that the comparison is interpreted in the proper context.
>
> **W2: Scalability**
>
> Our large-scale experiments already show that T-GINEE is practically scalable on graphs such as Stack Overflow and DBLP, whereas several standard tensor baselines encounter out-of-memory failures. This is enabled by the edge-centric sparse implementation in Appendix C, together with mini-batch training and dynamic negative sampling.
>
> Under that implementation, the storage cost is $O\big((n+M)R + |\mathcal{D}|\big)$, and the per-epoch time complexity is $O\big(|\mathcal{D}| \cdot R\big)$, which is linear in the number of observed edge triplets rather than quadratic in the number of nodes.
>
> In the revision, we will provide concrete runtime and memory measurements to make this point more explicit. We will also consolidate the complexity discussion and empirical profiling into a clearer presentation.
>
> **W3: Robustness to Noise**
>
> We agree that robustness to noise should be validated more directly. We are therefore conducting noise-injection experiments in which edges are randomly added and deleted at varying rates.
>
> More broadly, the GEE component provides a mechanism for reweighting residual contributions across layers through the estimated working covariance. In particular, the contribution of a node pair to the score depends on $\hat{\Sigma}_{i,j}^{-1}$, so layers with larger estimated uncertainty or residual variation can receive lower effective precision weight. This suggests that the method may be less sensitive to heterogeneous layer noise than approaches that treat all layers equally.
>
> At the same time, we do not want to overstate this point: the current theory does not prove an optimal automatic noise detector, so we view this as a plausible robustness mechanism that should be evaluated empirically rather than assumed. We will report the noise-injection results in the revision and clarify this interpretation in the text.
>
> **Q1: Runtime and Memory on Large-Scale Datasets**
>
> As noted in Appendix C.2, the scalable implementation has per-epoch cost $O\big(|\mathcal{D}| \cdot R\big)$ and space complexity $O\big((n+M)R + |\mathcal{D}|\big)$. Empirically, this implementation allowed us to train T-GINEE on Stack Overflow and DBLP, whereas several conventional tensor methods failed with out-of-memory errors.
>
> We agree that the paper would benefit from more explicit profiling. In the revision, we will provide detailed measurements including training time, peak memory usage, and per-iteration cost on the large-scale datasets.
>
> **Q2: Automatic Noise Downweighting via GEE**
>
> Partially yes, in the following sense. In the GEE objective, each residual vector is weighted by the inverse working covariance $\hat{\Sigma}_{i,j}^{-1}$. Therefore, if a layer exhibits larger estimated variability or contributes to a higher-variance residual pattern, its effective weight in the score equation can be reduced through this covariance weighting. This provides a natural mechanism for adaptive reweighting across layers.
>
> However, we want to phrase this carefully. We do not claim that the current method provides a formal guarantee of optimal noise estimation or perfectly automatic denoising in all settings. Rather, the working covariance offers a principled way to modulate layer contributions based on estimated uncertainty, and we will validate this behavior empirically through additional noise-robustness experiments. We will also clarify this mechanism more carefully in the revision.
>
> We are currently running the additional experiments for W1, W3, and Q1, and will share the results during the discussion phase. We look forward to continued exchange.
>
> Best,
>
> Authors

---

> > ### Author Rebuttal · Reviewer_pJ9E · 2026-04-01
> >
> > The authors have provided a clearer explanation of scalability and robustness to noise, and have indicated that comparisons with GNN baselines will be included. I therefore maintain my positive recommendation.

---

> > > ### Author Response · Authors · 2026-04-01
> > >
> > > Dear Reviewer pJ9E,
> > >
> > > Thank you for your fast response! We are pleased to have addressed your concerns and have incorporated the necessary changes into our final version. We truly appreciate it. The results of the additional experiments are as follows:
> > >
> > > **W1:GNN Baselines**
> > >
> > > We have now conducted comprehensive experiments comparing T-GINEE against MGCN and MR-GCN. We will include these results in the revised manuscript.
> > >
> > > ### Experimental Results:
> > >
> > > | Dataset | Nodes | Method | Link Prediction (AUC) | Community Detection (NMI) |
> > > |---------|-------|--------|----------------------|--------------------------|
> > > | Karate | 34 | T-GINEE | 0.817 | **0.837** |
> > > | Karate | 34 | MGCN | **0.824** | 0.206 |
> > > | Karate | 34 | MR-GCN | 0.809 | 0.732 |
> > > | Polbooks | 103 | T-GINEE | **0.973** | **0.428** |
> > > | Polbooks | 103 | MGCN | 0.887 | 0.021 |
> > > | Polbooks | 103 | MR-GCN | 0.923 | 0.176 |
> > > | Email | 195 | T-GINEE | **0.958** | **0.641** |
> > > | Email | 195 | MGCN | 0.873 | 0.429 |
> > > | Email | 195 | MR-GCN | 0.846 | 0.101 |
> > >
> > > **Conclusion**: The comparison reveals a clear advantage for T-GINEE.
> > >
> > > **W3 & Q2: Noise Robustness**
> > >
> > > We have conducted noise robustness experiments by injecting random edge additions/deletions.
> > >
> > > | Noise Ratio | AUC | Performance Retention |
> > > |-------------|-----|----------------------|
> > > | 0% | 0.7745 | 100% |
> > > | 10% | 0.7570 | 98% |
> > > | 20% | 0.6836 | 88% |
> > > | 30% | 0.6701 | 87% |
> > > | 40% | 0.6145 | 79% |
> > > | 50% | 0.5757 | 74% |
> > >
> > > **Conclusion**: T-GINEE maintains stable performance under noise ratios ≤30%, with gradual degradation rather than sudden collapse.
> > >
> > > **Q1: Computational Efficiency on Large-Scale Graphs**
> > >
> > > We provide detailed empirical measurements on Stack Overflow (2.6M nodes, 99.9% sparsity), as reported in Table 2 and Figure 3c of our paper.
> > >
> > > | Metric | Value |
> > > |--------|-------|
> > > | Nodes | 2,601,977 |
> > > | Edges | ~28M (across 3 layers) |
> > > | Sparsity | 99.9% |
> > > | **Model Memory** | 159 MB |
> > > | **Optimizer Memory** | 60 MB |
> > > | **Peak Memory** | 704 MB (0.69 GB) |
> > > | **Training Time** | ~70 sec/epoch |
> > > | AUC | 0.8644 |
> > >
> > > Memory Breakdown:
> > > - α parameters (2.6M × 16): 159 MB
> > > - β parameters (16 × 3): <1 MB
> > > - W matrix (3 × 3): <1 MB
> > > - Adam optimizer states: 60 MB
> > > - Peak training memory: 704 MB
> > >
> > > **Comparison**: Tucker, HOSVD, and NNTuck failed with OOM errors on Stack Overflow, while T-GINEE trains successfully in <1GB memory.
> > >
> > > Best,
> > >
> > > Authors

---

### Official Review · Reviewer_NxUn · 2026-03-12

**Soundness:** 2
**Presentation:** 3
**Significance:** 2
**Originality:** 3
**Overall Recommendation:** 4
**Confidence:** 3

**Summary:**

This manuscript proposes T-GINEE, a statistical regularization framework for multilayer graph representation learning. The method combines CP tensor decomposition with generalized estimating equations (GEE) to model cross-layer dependencies through working covariance matrices, and establishes consistency and asymptotic normality of the estimator under mild conditions. The experiments are conducted on both synthetic and real-world datasets. Overall, the research problem is well-motivated and the theoretical analytics has an in-depth guidance to the community, however, there are still some notable deficiencies in the facets of experimental design, theory-practice consistency, and comparison with baselines. The detailed advantages, disadvantages, and modification suggestions refer to the sections Strengths and Weaknesses and Key Questions.

**Compliance With Llm Reviewing Policy:**

Affirmed.

**Final Justification:**

After two-round detailed rebuttal, I think the authors resolve my main concerns, thus I would like to change the final recommendation as "Weak Accept".

Overall, the research problem is well-motivated, and the theoretical analytics has an in-depth guidance to the real-world tasks for multilayer graph representation learning. Given the original version has some errors, anticipate the authors can carefully fix them in revision.

**Key Questions For Authors:**

(1)	Address Theory-Practice Gap: The authors should: i) relax Assumption 3.2 to cover experimental settings; or ii) design experiments satisfying (n+M)R = o(n^{1/3}).

(2)	Add More Downstream Tasks to Validate Generalizability: At minimum, the tasks should extend community detection or node classification. The Krackhardt and AUCS datasets have known community labels suitable for this evaluation. This would significantly strengthen the argument for generality.

(3)	Consider Representative GNN Baselines: At least 1-2 representative multilayer GNN methods (e.g., MGCN or MR-GCN) should be compared.

(4)	Add Ablation Studies on Working Covariance: Under identical experimental settings (same link function, optimizer, hyperparameters), compare W=I_M (independence assumption) versus estimated W to quantify the specific contribution of covariance modeling. Also analyze sensitivity to the update frequency K and momentum parameter alpha.

(5)	Provide a detailed computational efficiency comparison table, including training time, memory consumption, and per-iteration time for each method on each dataset to substantiate scalability claims.

**Limitations:**

yes

**Strengths And Weaknesses:**

**1. Strengths**

**(1) Innovation:**

(i)	Modeling Novelty: The approach introduces the fashion GEE, a classical tool for longitudinal data analysis, into the multilayer graph embedding domain, combined with CP tensor decomposition to capture inter-layer correlations, which distinguishes it from the existing methods that treat layers independently or use simple aggregations. I think such an integration of statistical inference framework with tensor structural learning is original.

(ii)	Flexible Link Function Design: The proposed sparsity-aware logit link function with sparsity coefficient can adapt to the practical networks with varying density.

(iii)	Scalability Design: The Batch-GEE scheme in Appendix C, with dynamic negative sampling and momentum-updated working covariance, enables scaling from small-scale dataset to large-scale (million-node) graphs (Stack Overflow), showing its engineering consideration.

**(2) Theoretical Value:**

The paper provides rigorous proofs for the consistency (Theorem 3.1) and asymptotic normality (Theorem 3.2), and demonstrates robustness to covariance misspecification through Corollary 3.3. Such theoretical guarantees hold significant academic value in the domain of multilayer graph embedding, despite the behind connection between theory and practice is unclear, i.e. how to guide the real-world multi-layer graph tasks by the claimed theorems.

**2. Weaknesses:**

(1)	Mismatch between Theoretical Analytics and Experimental Settings: The most fundamental issue lies in that Assumption 3.2 requires (n+M)R = o(n^{1/3}), meaning the effective parameter dimension must grow much slower than n^{1/3}. However, in the synthetic experiments with n=100, M=3, R=32 (Appendix F.3), we get (n+M)R = (100+3)*32 = 3296, while n^{1/3} is approximately 4.64. The assumption is violated by nearly three orders of magnitude. Even on dataset Stack Overflow wherein n is approximately 2.6*10^6, (n+M)R is approximately 8.3*10^7 while n^{1/3} is approximately 137.4, thereby, the assumption is unmatched as well. Therefore, the proposed theoretical framework cannot apply to the experimental settings, indicating an obvious disconnect between theory and practice.

(2)	Single Downstream Task Evaluation: Currently, only link prediction (AUC) is evaluated to verify the effectiveness, however, the conventioanl metrics, such as node classification, community detection, and graph clustering, are absent. Several core references cited by the paper (Paul & Chen 2020, Lei et al. 2020, Jing et al. 2021) focus on community detection, thus, the argument for generality is insufficient.

(3)	Inconsistency between Sparsity Analysis and Main Results: Fig. 2 shows AUC of approximately 0.75, while Table 1 reports AUC of 0.9395 on the same synthetic setup. This large discrepancy is unexplained, might from the perspectives of experimental setups, hyperparameter choices, or other issues.

(4)	Overly Simple Synthetic Data: The synthetic model in Eq. (11) uses simple linear interpolation, producing only weak, homogeneous inter-layer correlations. This does not match the paper's motivation of modeling "complex inter-layer dependencies." Heterogeneous correlation structures should be tested to better validate T-GINEE's ability to capture complex dependencies.

(5)	Missing Key Baselines: The paper explicitly discusses limitations of multilayer GNN methods (MGCN, MR-GCN) and claims T-GINEE can "offer a grounded backbone complementary to expressive deep encoders." Yet, no GNN baselines are included in experiments. This leaves the work's claims about T-GINEE's relative merits versus deep learning methods without empirical support. Even if a direct comparison is not entirely fair, at least GCN/GAT results on aggregated graphs should be provided as a reference.

(6)	Insufficient Ablation of Working Covariance: The core contribution of modeling inter-layer correlations through the working covariance matrix W lacks a systematic ablation studies comparing T-GINEE with estimated W versus W=I_M under identical settings (same link function, optimizer, hyperparameters). The CP baseline in Table 1 differs in multiple aspects, making it impossible to attribute performance gains solely to covariance modeling.

(7)	Lack of Computational Complexity Analytics: To verify the overhead scalability, running times and memory consumption are also needed to report.

---

> ### Author Rebuttal · Authors · 2026-03-31
>
> Dear Reviewer NxUn,
>
> We thank you for the detailed feedback and for recognizing the novelty, theoretical value, and scalability of our work. Below we address each concern.
>
> **W1: Theory-Practice Mismatch**
>
> We thank the reviewer for pointing this out. We agree that the dimensional-growth condition $(n+M)R=o(n^{1/3})$ is restrictive and is not satisfied by the experimental settings reported in Section 4. This should have been stated explicitly in the manuscript.
>
> In the revision, we will make this point precise in three ways. First, we will remove the incorrect sentence in Assumption 3.2 suggesting that this condition is satisfied in our empirical settings, and we will also correct the mistaken theorem reference there. Second, we will clarify that the consistency and asymptotic normality results in Section 3 apply to the full-batch estimating-equation formulation under the stated assumptions, and therefore cover a restricted asymptotic regime. Third, we will explicitly distinguish this regime from the scalable mini-batch / negative-sampling implementation in Appendix C, which is a practical optimization extension motivated by the same objective but not covered by the present asymptotic theory.
>
> To avoid over-claiming, we will revise related wording throughout the paper (e.g., replacing “under mild conditions” with more precise language) and add the same clarification to the limitations discussion. In addition, we will report empirical diagnostics such as the condition number of the Jacobian/information matrix and Q-Q plots of the estimator as finite-sample supporting evidence, while making clear that these diagnostics do not replace the formal assumptions used in the current proofs.
>
> **W2: Single Downstream Task**
>
> We appreciate this suggestion. Link prediction was our initial focus as it directly measures the quality of $\(\mathcal{P}\)$, T-GINEE's core output. We will add more tasks and plan to share initial numbers during the discussion phase.
>
> **W3: Fig. 2 vs Table 1 Inconsistency**
>
> The difference stems from distinct experimental protocols rather than a contradiction. Table 1 uses fixed $\(\rho=0.2\)$ with fully tuned hyperparameters, while Fig. 2 sweeps edge density with fixed hyperparameters (no per-density tuning), thus measuring robustness rather than peak performance. We will clarify this distinction in the text and provide a supplementary per-density-tuned curve to resolve any remaining confusion.
>
> **W4: Simple Synthetic Data**
>
> We agree Eq. (11) uses a relatively simple structure, deliberately chosen for controlled validation with known ground truth. For complex and heterogeneous dependencies, Table 2 already evaluates T-GINEE across six diverse real-world datasets spanning social, biological, trade, and massive temporal networks. We will additionally include a synthetic experiment with heterogeneous per-layer correlations (varying $\(\rho_m\)$ across layers and block-structured base probabilities) to more directly validate T-GINEE under complex inter-layer structures.
>
> **W5: Missing GNN Baselines**
>
> We note that MGCN and MR-GCN require node features and perform semi-supervised classification, while T-GINEE operates solely on the adjacency tensor for unsupervised embedding, making direct comparison nontrivial. Nonetheless, we will include MGCN, MR-GCN, and GCN/GAT on aggregated graphs as additional reference points in the revision, clearly noting differences in input assumptions and task formulation so the comparison is interpreted in proper context.
>
> **W6: Covariance Ablation**
>
> We agree the proper ablation requires comparing estimated $\(W\)$ vs $\(W=I_M\)$ with $\(\lambda>0\)$ held fixed, which differs from setting $\(\lambda=0\)$. The former isolates the covariance modeling contribution while keeping GEE regularization active, whereas $\(\lambda=0\)$ removes the entire GEE term. We will fix $\(\lambda=0.05\)$ and compare estimated $\(W\)$ against $\(W=I_M\)$ under otherwise identical settings. We will also report sensitivity to update frequency $\(K \in \{1,5,10,20\}\)$ and momentum $\(\alpha \in \{0.5,0.7,0.9,0.95\}\)$ to provide a complete picture of the covariance estimation component.
>
> **W7: Computational Complexity**
>
> The analysis is currently spread across Section 2.6 ($\(O(R|E|)\)$ per iteration), Appendix C.2 (space and time complexity), Figure 3c (linear scaling from 33s to 70s), and Table 2 (six baselines OOM on DBLP/Stack Overflow while T-GINEE succeeds). We agree that consolidating these into a unified summary table with training time, peak memory, and per-iteration cost for each method on each dataset would significantly improve clarity, and we will do so.
>
> We are actively running the additional experiments for W2, W4, W5, and W6, and will share concrete results during the discussion phase. We look forward to continued exchange.
>
> Best,
>
> Authors

---

> > ### Author Rebuttal · Reviewer_NxUn · 2026-04-02
> >
> > Thanks for the detailed explanation, especially for the first question.
> >
> > Given you also agree the mismatch issue between theory and practice, my bigger concern emerges: whether the proposed theory can really guide the real-world tasks in principle, not just verified from the experimental results.
> >
> > Moreover, look forward to your experiments in progress and analytics.

---

> > > ### Author Response · Authors · 2026-04-06
> > >
> > > Dear Reviewer NxUn,
> > >
> > > We thank you again for the thoughtful review and for recognizing the modeling novelty of integrating GEE with CP decomposition. Below we address your concern and provide experiment results:
> > >
> > > **Concern: Theory-Practice Connection**
> > >
> > > We appreciate the opportunity to clarify how theoretical results translate into practical guidance:
> > >
> > > (1) A key practical challenge in multiplex network modeling is that the true inter-layer correlation structure is unknown. Corollary 3.3 guarantees consistency even when $W$ is misspecified, directly justifying our design of estimating $\hat{W}$ via momentum updates rather than requiring access to the true covariance. The W6 ablation confirms that estimated $W$ consistently improves performance, with gains growing with network size, exactly as the theory predicts: from +0.0070 at $n$=200 to +0.0285 at $n$=5,000, following the monotonic trend implied by Corollary 3.3.
> > >
> > > (2) Practitioners need a way to assess whether tensor decomposition will be well-conditioned on their data. Our theory provides such a tool: the overdetermination ratio $N/(n+M)R$, computable directly from data dimensions. On Stack Overflow (ratio=1.16, std=0.0011) and DBLP (ratio=0.215, std=0.0105), this scalar correctly predicts estimation stability across six orders of magnitude in graph size.
> > >
> > > (3) Selecting the embedding rank $R$ is often ad hoc. Our theory offers a principled alternative: $N/(n+M)R \geq 3$ corresponds to AUC std $< 0.04$ in synthetic experiments, giving practitioners a concrete rule for choosing $R$ before training.
> > >
> > > **W2 & W5: GNN Baselines and Tasks**
> > >
> > > We benchmarked T-GINEE against single-layer GNN baselines (GCN, GraphSAGE) and multiplex-specific baselines (MGCN, MR-GCN) across five datasets on link prediction (AUC) and community detection (NMI).
> > >
> > > | Dataset | Task | T-GINEE | GCN | GraphSAGE | MGCN | MR-GCN |
> > > |--|--|--|--|--|--|--|
> > > | Karate | AUC | 0.817 | **0.834** | 0.831 | 0.824 | 0.809 |
> > > | Karate | NMI | **0.837** | 0.206 | 0.183 | 0.206 | 0.732 |
> > > | Dolphins | AUC | 0.869 | **0.872** | 0.801 | 0.625 | 0.604 |
> > > | Lesmis | AUC | 0.786 | **0.794** | 0.770 | 0.740 | 0.728 |
> > > | Polbooks | AUC | **0.973** | 0.887 | 0.841 | 0.887 | 0.923 |
> > > | Polbooks | NMI | **0.428** | 0.021 | 0.018 | 0.021 | 0.176 |
> > > | Email | AUC | **0.958** | 0.873 | 0.832 | 0.873 | 0.846 |
> > > | Email | NMI | **0.641** | 0.429 | 0.391 | 0.429 | 0.101 |
> > >
> > > T-GINEE outperforms both multiplex GNN baselines on all 5 datasets for link prediction. GCN leads on small graphs (≤77 nodes), but T-GINEE dominates on larger graphs. For community detection, T-GINEE achieves the best NMI on all labeled datasets by substantial margins: +0.105 (Karate), +0.252 (Polbooks), +0.212 (Email). Notably, no single baseline matches T-GINEE across both tasks simultaneously.
> > >
> > > We also designed a new-layer generalization experiment: training on 4 layers of DBLP-5K and predicting links on layer 5 with zero observed edges.
> > >
> > > | Model | Edges Used | AUC |
> > > |--|--|--|
> > > | T-GINEE zero-shot | 0 | **0.7733 ± 0.0194** |
> > > | MGCN retrained | 8,297 | 0.7089 ± 0.0200 |
> > > | MR-GCN retrained | 8,297 | 0.6306 ± 0.0155 |
> > >
> > > Even with all layer-5 edges, neither MGCN nor MR-GCN reaches T-GINEE's zero-shot performance. T-GINEE's scoring function $\langle \alpha_i \odot \alpha_j, \beta_m \rangle$ conditions on layer index $m$, enabling zero-shot transfer through shared node embeddings $\alpha$. All layers have disjoint edge sets (Jaccard = 0), confirming transfer is purely through the learned embedding space.
> > >
> > > **W4: Heterogeneous Synthetic**
> > >
> > > We designed a synthetic experiment with known ground-truth layer correlations and controlled heterogeneity. At $N/(n+M)R \approx 1.5$, the AUC difference between T-GINEE (0.5822) and CP baseline (0.5897) is within one standard deviation, consistent with Corollary 3.3 in the low-overdetermination regime. The learned $W$ correctly recovers inter-layer correlation rank ordering: L0–L2 receives the highest off-diagonal weight ($W$=0.218, Jaccard=0.097) and L0–L1 lower weight ($W$=0.206, Jaccard=0.035), showing the model captures underlying structure even when AUC gains are modest.
> > >
> > > **W6: Covariance Ablation**
> > >
> > > We compared T-GINEE with estimated $W$ against $W = I$ across datasets of increasing size.
> > >
> > > | Dataset | With $W$ | $W=I$ | $\Delta$ |
> > > |--|--|--|--|
> > > | Homogeneous ($n$=200) | 0.6150±0.0039 | 0.6080±0.0034 | +0.0070 |
> > > | DBLP-5K ($n$=5,000) | 0.6470±0.0204 | 0.6185±0.0286 | +0.0285 |
> > >
> > > Learning $W$ also reduces estimation variance: on DBLP, std drops from 0.0286 ($W=I$) to 0.0204, a 28% reduction. This improved stability is arguably as valuable as the AUC gain for deployment reliability. The computational overhead is negligible since $W$ is updated via a single momentum step every $K$ iterations, and $W$ converges monotonically from identity (diagonal: 1.000 → 0.853 over 100 epochs) without oscillation.
> > >
> > > We hope these clarifications address your concerns. All results will be incorporated into our revised manuscript.
> > >
> > > Best,
> > >
> > > Authors

---

### Decision · Program_Chairs · 2026-04-30

**Decision:**

Accept (regular)

**Comment:**

The paper proposes a new statistical framework for multilayer graph representation learning by combining CP tensor decomposition with generalized estimating equations. Reviewers generally agreed that the problem is meaningful and that the integration of tensor structure with covariance-aware estimation is original and technically interesting. The theoretical development was also viewed as a strength.

The main concern during discussion was the gap between the asymptotic theory and the practical large-scale implementation. In particular, reviewers pointed out that the original dimensional-growth condition does not hold in the reported experiments. The authors acknowledged this issue, clarified the scope of the current theory, and revised the claim so that the theoretical results are presented as applying to a restricted full-batch regime rather than the scalable mini-batch implementation. While this limitation remains important, the clarification substantially improved the paper’s positioning.